# TRP14 is the rate-limiting enzyme for intracellular cystine reduction and regulates proteome cysteinylation

Pablo Martí-Andrés [iD] [1,2], Isabela Finamor[1,3], Isabel Torres-Cuevas [iD] [1], Salvador Pérez [iD] [1],
Sergio Rius-Pérez [iD] [1], Hildegard Colino-Lage [4], David Guerrero-Gómez [iD] [4], Esperanza Morato [iD] [5],
Anabel Marina[5,6], Patrycja Michalska[7], Rafael León[8], Qing Cheng[2], Eszter Petra Jurányi [iD] [9,10],
Klaudia Borbényi-Galambos[9,11], Iván Millán[12], Péter Nagy[9,13,14], Antonio Miranda-Vizuete [iD] [4],
Edward E Schmidt [iD] [13,15], Antonio Martínez-Ruiz[16], Elias SJ Arnér [iD] [2,17✉] & Juan Sastre [iD] [1✉]

## Abstract

It has remained unknown how cells reduce cystine taken up from the extracellular space, which is a required step for further utilization of cysteine in key processes such as protein or glutathione synthesis. Here, we show that the thioredoxin-related protein of 14 kDa (TRP14, encoded by *TXNDC17*) is the rate-limiting enzyme for intracellular cystine reduction. When TRP14 is genetically knocked out, cysteine synthesis through the transsulfuration pathway becomes the major source of cysteine in human cells, and knockout of both pathways becomes lethal in *C. elegans* subjected to proteotoxic stress. TRP14 can also reduce cysteinyl moieties on proteins, rescuing their activities as here shown with cysteinylated peroxiredoxin 2. *Txndc17* knockout mice were, surprisingly, protected in an acute pancreatitis model, concomitant with activation of Nrf2-driven antioxidant pathways and upregulation of transsulfuration. We conclude that TRP14 is the evolutionarily conserved enzyme principally responsible for intracellular cystine reduction in *C. elegans*, mice, and humans.

**Keywords** Cysteine Homeostasis; Protein Cysteinylation; Transsulfuration; Acute Pancreatitis; Proteotoxic Stress
**Subject Categories** Metabolism; Post-translational Modifications & Proteolysis

## Introduction

Cystine is taken up by cells through the $x_C^-$ cystine/glutamate antiporter system (SLC7A11), also known as xCT transporter, as a source of cysteine that complements its synthesis from methionine through the transsulfuration pathway (McBean, 2012; Garg et al, 2011). In cells requiring much cysteine, such as many tumor cells or upon states of oxidative stress, the $x_C^-$-mediated uptake of cystine with its further reduction into two molecules of cysteine becomes particularly important (Arensman et al, 2019). It has yet remained unknown how the intracellular reduction of cystine is catalyzed, although both the thioredoxin (Trx) and glutathione (GSH) systems have been implicated (Paul et al, 2018).

Cysteine residues in proteins can be easily oxidized, with the enzymatic control of reversible disulfide bond oxidation on the sulfur of cysteine being important steps in redox signaling pathways and control of protein function (Lo Conte and Carroll, 2013; Arnér and Holmgren, 2000). Oxidative stress, such as occurring during ischemia-reperfusion injuries, distorted aerobic metabolism, proteostatic collapse, or inflammatory oxidative bursts, is a state promoting oxidative damage of cellular components including the oxidation of protein thiols, leading to the formation of protein disulfides or mixed disulfides between proteins and low-molecular-weight thiols, like cysteine, GSH and γ-glutamylcysteine (Eaton, 2006; Moreno et al, 2014). The Trx and GSH systems are critical enzymatic systems in controlling the redox status of cysteine residues in proteins through reductive catalysis. The Trx system includes isoforms of Trx-fold proteins and thioredoxin reductase (TrxR), with the latter using NAPDH to reduce the active site disulfide

[1]Department of Physiology, Faculty of Pharmacy, University of Valencia, Burjassot, Valencia, Spain. [2]Division of Biochemistry, Department of Medical Biochemistry and Biophysics, Karolinska Institutet, SE 171 77 Stockholm, Sweden. [3]Department of Physiology and Pharmacology, Federal University of Santa Maria, Santa Maria, Rio Grande do Sul, Brazil. [4]Redox Homeostasis Group, Instituto de Biomedicina de Sevilla (IBIS), Hospital Universitario Virgen del Rocío/CSIC/Universidad de Sevilla, Seville, Spain. [5]Centro de Biología Molecular "Severo Ochoa" (CBMSO), CSIC-UAM, Madrid, Spain. [6]Unidad de Técnicas Bioanalíticas (BAT), Instituto de Investigación de Ciencias de la Alimentación (CIAL), CSIC-UAM, Madrid, Spain. [7]Department of Chemistry, Molecular Sciences Research Hub, Imperial College London, London, UK. [8]Institute of Medical Chemistry, CSIC, Madrid, Spain. [9]Department of Molecular Immunology and Toxicology and the National Tumor Biology Laboratory, National Institute of Oncology, Budapest, Hungary. [10]Molecular Medicine Division, Semmelweis University Doctoral College, Budapest, Hungary. [11]Kálmán Laki Doctoral School, University of Debrecen, Debrecen, Hungary. [12]Instituto Cavanilles de Biodiversidad y Biología Evolutiva, Universidad de Valencia, Paterna, Valencia, Spain. [13]Department of Anatomy and Histology, HUN-REN-UVMB Laboratory of Redox Biology, University of Veterinary Medicine, Budapest, Hungary. [14]Chemistry Institute, University of Debrecen, Debrecen, Hungary. [15]Department of Microbiology and Cell Biology, Montana State University, Bozeman, MT, USA. [16]Unidad de Investigación, Hospital Universitario Santa Cristina, Instituto de Investigación Sanitaria Princesa (IIS_IP), Madrid, Spain. [17]Department of Selenoprotein Research and the National Tumor Biology Laboratory, National Insitute of Oncology, Budapest, Hungary.
✉E-mail: elias.arner@ki.se; juan.sastre@uv.es

of Trx substrates to a dithiol, further channeling the reducing equivalents to downstream targets (Arnér and Holmgren, 2000).

The thioredoxin-related protein of 14 kDa (TRP14, encoded by the *TXNDC17* gene in humans) is a Trx-fold protein with the active-site sequence WCPDC (Jeong et al, 2004b) which, in having an acidic aspartate (D) residue between the redox-active cysteine (C) moieties, is unique in the thioredoxin family of proteins yet is highly conserved across phyla. TRP14 is an ubiquitously expressed cytosolic oxidoreductase that is efficiently reduced by cytosolic TrxR1 and was initially found to suppress NF-κB signaling (Jeong et al, 2004a; Espinosa and Arnér, 2019). Interestingly, TRP14 is unable to reduce typical Trx1 protein substrates, such as insulin, ribonucleotide reductase, peroxiredoxins (Prxs), and methionine sulfoxide reductase, suggesting that TRP14 may have other specific functions compared to Trx1 (Jeong et al, 2004b; Espinosa and Arnér, 2019). Using pure enzyme systems we previously found that TRP14 is more efficient than Trx1 in reducing the disulfide of cystine, thus producing two cysteine molecules in a reaction driven by TrxR1 and NADPH (Pader et al, 2014). We also found that TRP14 reactivates oxidized protein tyrosine phosphatase PTP1B (Dagnell et al, 2013), and efficiently reduces persulfidated (Dóka et al, 2016, 2020) and nitrosylated (Pader et al, 2014) cysteine residues in proteins.

Although TRP14 can reduce cystine in vitro (Pader et al, 2014) it is unknown whether TRP14 has this role in vivo, and it should be noted that although several Trx-fold proteins can catalyze reduction of cystine when directly assayed, this does not necessarily suggest they have that role in a physiological cell context. Preliminary results of ours suggested that TRP14 could affect both intracellular signaling pathways and cystine utilization in HEK293 cells, but the results were inconclusive (Espinosa Fernández B, 2020). It is also not known if TRP14 can reduce the disulfide motif found in cysteinylated proteins. Our present study reveals that TRP14 is the hitherto unknown rate-limiting reductase for cytosolic conversion of cystine into cysteine, and that it can reduce cysteinylated proteins. The physiological impact of these activities becomes evident under various oxidative stress conditions, as here shown in *C. elegans*, human cells, and mouse models. The interplay revealed here between TRP14 and the transsulfuration pathway as sources of cysteine, shown in metazoan model systems including nematodes, mice, and human, suggests cystine reduction, at least in part, underlies the evolutionary conservation of this odd Trx family member that has yet not been understood. Moreover, the genetic tools and established stress protocols afforded by the *C. elegans* and mouse models allowed in vivo assessment of the roles of TRP14 under conditions of proteotoxic stress or acute pancreatitis, respectively. The results show that while TRP14 upholds intracellular cystine reduction and decysteinylation of proteins under normal conditions, its knockout in combination with oxidative stress leads to accentuated activation of Nrf2 and upregulation of the transsulfuration pathway that result in an, at first seemingly paradoxical, protection against cell or tissue damage.

## Results

### Deletion of TRP14 in human HEK293 cells severely hampers their capacity to reduce cystine, but induces the transsulfuration pathway and activates Nrf2 upon exogenous challenge with cystine

In human TRP14 knockout HEK293 cells, protein levels of Trx1 were upregulated (Fig. 1A), and cysteine levels were lower than in parental HEK293 cells (Appendix Fig. S1A). The upregulation of Trx1, however, did not seem to compensate fully for the loss of TRP14 activities, because when assessing cystine reduction using BODIPY-labeled cystine, which becomes fluorescent upon reduction of its disulfide (Pader et al, 2014), TRP14 deletion severely hampered the strong fluorescence seen in control cells (Fig. 1B). Kinetic assays confirmed the markedly decreased cystine reducing capacity of TRP14 KO cells (Fig. 1C), which was recovered when overexpressing a wild-type version of the enzyme in TRP14 KO cells, but not when overexpressing an active site Cys-to-Ser mutant version of the enzyme (Fig. 1D,E). Furthermore, the cystine reduction capacity of parental HEK293 cells overexpressing wild-type TRP14 was higher than that of HEK293 cells overexpressing the active-site mutant version (Fig. 1D,E). These observations suggested that TRP14 could be the rate-limiting enzyme for intracellular cystine reduction and that this activity could not be fulfilled by Trx1 alone within the cellular context.

Next, we considered whether the transsulfuration pathway could be involved in providing cysteine when TRP14 is absent by using a fluxomic approach with heavy sulfur ($[^{34}S]$)-labeled methionine to follow the incorporation of the labeled sulfur into sulfur-containing metabolites in WT and TRP14 KO cells. Whereas typical cell culture media contains super-physiological levels of cystine (200 μM), we here mimicked in vivo conditions using 4 μM cystine in the culture medium. Results showed larger proportions as well as elevated steady-state levels of heavy cystathionine, heavy cysteine and heavy reduced glutathione (GSH) in TRP14 KO cells compared to their WT counterparts (Fig. 2A–C), indicating that the transsulfuration pathway is more active in TRP14 KO cells and, conversely, that in WT cells a larger proportion of cysteine is likely to come from cystine through the $x_C^-$ system with subsequent TRP14-mediated reduction. These data suggested that in the absence of TRP14, the transsulfuration pathway provides an increased source of cysteine synthesized from methionine-derived homocysteine.

Interestingly, using the super-physiological concentration of cystine in standard culture media (200 μM), we found no differences in cystathionine flux between TRP14 KO and WT cells, and cysteine was not significantly produced by the transsulfuration pathway (Appendix Fig. S1B–D; see "Discussion").

TRP14 knockout cells were significantly more resistant than control cells when exposed to the oxidative stress triggered by 500 μM cystine (Fig. 3A). We reasoned that Nrf2 activation upon exposure to cystine (Liu et al, 2020) might have conferred this resistance of TRP14 knockout cells. This was therefore assessed next, using our previously developed pTRAF tool that reports upon transcription factor activities using fluorescence (Johansson et al, 2017; Kipp et al, 2017). Nrf2 activation was indeed markedly higher in TRP14 knockout cells exposed to cystine as compared to controls, while both cell types activated Nrf2 upon treatment with the strong Nrf2 activator auranofin (Fig. 3B,C). NF-κB activation upon stimulation with TNF-α was lower in TRP14 knockout cells than in controls, while the HIF-1α activity was consistently low in all tested conditions (Fig. 3B,C).

### Transsulfuration and TRP14-catalyzed cystine reduction pathways are essential for *Caenorhabditis elegans* survival upon proteotoxic stress

Wishing to explore whether cystine reduction by TRP14 would be an evolutionarily conserved pathway, we next turned to the *C. elegans*

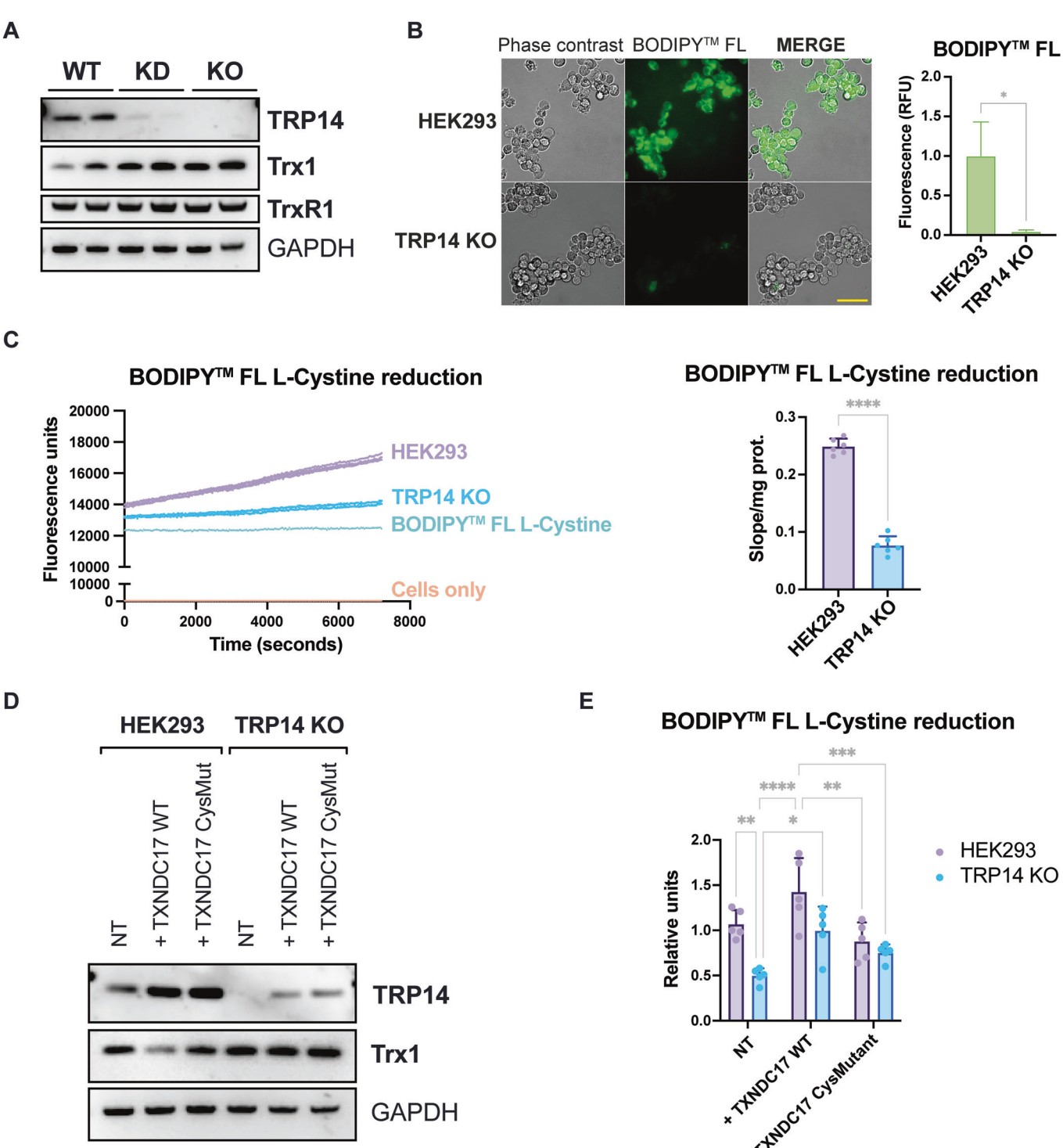

model. This nematode has a well-characterized thioredoxin system, highly similar to that of mammals (Johnston and Ebert, 2012), including a TRP14 orthologue (TXDC-17). Like all metazoans, *C. elegans* also has a conserved transsulfuration pathway that can provide cells with cysteine, where *cbs-1* and *cbs-2* genes encode two isoforms of cystathionine β-synthase (Vozdek et al, 2012) and *cth-1* and *cth-2* genes encode two isoforms of cystathionine γ-lyase (Qabazard et al, 2014).

First we confirmed that, like human TRP14 (Pader et al, 2014), the *C. elegans* TRP14 orthologue TXDC-17, together with its native reductase, could efficiently reduce cystine in vitro (Fig. 4A). To investigate the role of the *C. elegans* TRP14 orthologue in vivo, we generated a *txdc-17(syb4767)* null allele by CRISPR-Cas9 technology (Fig. EV1). Like *Txndc17* knockout in mammalian systems, *txdc-17(syb4767)* nematodes were also viable, grew normally and were indistinguishable from wild-

**Figure 1. TRP14 is the rate limiting enzyme for cystine reduction in HEK293 cells.**

(A) Representative western blot of TRP14, Trx1, and TrxR1 in HEK293 (WT) and TRP14 knockdown (KD) and knockout (KO) cells. (B) Representative fluorescence microscopy image showing BODIPY™ FL L-Cystine reduction as green fluorescence in HEK293 and TRP14 KO cells 15 min after the addition of the labeled cystine. (C) BODIPY™ FL L-Cystine reduction capacity of HEK293 and TRP14 KO measured as an increase of BODIPY™ fluorescence over time (2 h). (D) Representative western blot image showing TRP14 and Trx1 levels in HEK293 and TRP14 KO cells overexpressing wild-type TRP14 ( + TXNDC17 WT) or an active-site mutant version of the enzyme ( + TXNDC17 CysMut). (E) BODIPY™ FL L-Cystine reduction capacity of HEK293 and TRP14 KO cells overexpressing wild-type TRP14 ( + TXNDC17 WT) or an active-site mutant version of the enzyme ( + TXNDC17 CysMut). $n = 3$–6. Data information: Two-way ANOVA and Tukey's test for multiple comparisons: *$P < 0.05$; **$P < 0.01$; ***$P < 0.001$; ****$P < 0.0001$. The indicated "$n$" corresponds to the number of biological replicates. Data plotted corresponds to the mean (columns) ± standard deviation (error bars). Scale bar: 50 μm. Source data are available online for this figure.

type in size and movement (Fig. 4B; Appendix Fig. S2). To impair the worm transsulfuration pathway, we first generated a double mutant *cbs-1; cbs-2*. However, expanding previous results obtained with *cbs-1* RNAi (Vozdek et al, 2012), *cbs-1(gk5787)* mutants grew extremely slowly (Fig. 4C), probably due to the accumulation of toxic levels of homocysteine (Khare et al, 2009). Although *cbs-2* has been proposed to be a pseudogene (Vozdek et al, 2012), *cbs-1; cbs-2* double mutants were even smaller than *cbs-1* single mutants, suggesting CBS-2 may have some residual enzymatic activity (Fig. 4C). We therefore decided to impair the transsulfuration pathway at the level of cystathionine γ-lyase (cth). Single and double *cth-1; cth-2* mutants were viable and showed no phenotype (Fig. 4B). Notably, a *txdc-17; cth-1; cth-2* triple mutant was also viable with no discernable phenotype, suggesting that the TRP14-catalyzed cystine reduction together with transsulfuration is not required under basal growth conditions in *C. elegans*.

To study the importance of the two pathways under oxidative stress conditions, we used a *C. elegans* model that expresses the Q40::YFP aggregation-prone protein in muscle cells (Morley et al, 2002), which induces a proteostatic challenge with oxidative stress and makes Q40:YFP animals extremely sensitive to glutathione depletion (Kirstein et al, 2015; Guerrero-Gómez et al, 2019). Inhibiting separately either the transsulfuration or the TRP14-catalyzed cystine reduction pathways in nematodes expressing Q40:YFP did not cause any phenotype. However, simultaneous impairment of both pathways in Q40:YFP nematodes caused a fully penetrant lethal phenotype (Fig. 4B). Interestingly, the lethal phenotype was obtained only with *cth-2* mutants as *txdc-17; cth-1; Q40:yfp* animals were viable and indistinguishable from *Q40::yfp* worms (Fig. 4B). This result is in consonance with previous data demonstrating non-redundant functions between *C. elegans cth-1* and *cth-2* genes (Warnhoff and Ruvkun, 2019) and suggests a key role of *cth-2*, but not *cth-1*, in cysteine synthesis through the transsulfuration pathway.

Importantly, we obtained equivalent results using *C. elegans unc-52(e669su250)* mutants, which undergo progressive paralysis due to aggregation of a metastable version of the perlecan orthologue protein UNC-52 (Mackenzie et al, 1978), which are also highly sensitive to glutathione depletion (Guerrero-Gómez et al, 2019). We failed to isolate *txdc-17; cth-2; unc-52(e669su250)* triple mutants, while *txdc-17; cth-1; unc-52(e669su250)* animals were viable and grew similar to wild-type (Fig. 4B).

## TRP14 status affects the extent of protein cysteinylation, and TRP14 can reduce cysteinylated proteins, including cysteinylated Prx2

We next hypothesized that TRP14, apart from being the rate-limiting enzyme for intracellular cystine reduction, may also directly reduce the

disulfide in cysteinylation motifs on proteins. To test this we used TRP14 stable knockdown HEK293 cells (Pader et al, 2014; Dóka et al, 2016), which exhibited a profile for TRP14 and Trx1 levels similar to TRP14 knockout cells (Fig. 1A). TRP14 knockdown cells were incubated with 250 μM cysteine-BIO for 1 h, and streptavidin-HRP was then used to detect biotin moieties present in protein lysates using western blot as a measure of cysteine addition onto proteins. This protein biotin-labeling was increased upon TRP14 knockdown (Fig. 5A), showing that lowering TRP14 levels indeed promoted cellular protein cysteinylation. We next used the cell lysates of the TRP14 knockdown cells, that had first been labeled with 250 μM cysteine-BIO for 1 h, as a substrate for in vitro assays. Incubating these lysates with the components of the TRP14 system (20 μM TRP14, 1 μM TrxR1, and 1 mM NADPH) resulted in a clear decrease of protein cysteinylation (Fig. 5B), supporting that TRP14 together with TrxR1 and NADPH can indeed reduce cysteinylated proteins. Control experiments showed that all components in the assay were required for efficient decysteinylation to occur (Appendix Fig. S3C), as was the case for components of the Trx1 system (Appendix Fig. S3D).

Using the Cys-BIO labeling in streptavidin-mediated pulldown experiments, we next aimed to identify individual proteins that had been cysteinylated using mass spectrometry. This analysis identified 42 proteins with higher cysteinylation levels ($P < 0.05$) in TRP14 knockdown cells compared to controls, including several ribosomal proteins as well as cytosolic peroxiredoxins Prx1 and Prx2 (Fig. 6). We also found that Prx2 can be cysteinylated in vitro, at the catalytically resolving Cys172 residue (Fig. 5C); for peptide identification see the PRIDE partner repository with the dataset identifier PXD050368. This cysteinylation of Prx2 inhibited its catalytic function, while TRP14 could directly decysteinylate Prx2 and thus reactivate it for $H_2O_2$ removal, as supported by Trx1 (Fig. 5D,E). Both TRP14 and Trx1 could efficiently decysteinylate Prx2 in vitro. However, since Trx1 has a wider substrate specificity than TRP14 it may be possible that Trx1 cannot keep up with decysteinylation of Prx2 in a more complex cellular setting; such scenario could be simulated in vitro using addition of insulin, that is reduced by Trx1 but not TRP14, which inhibited Prx2 decysteinylation by Trx1 but not by TRP14 (Fig. 5F). The decysteinylation of Prx2 by Trx1 was also strongly suppressed by addition of $H_2O_2$, which was not the case with TRP14 (Fig. 5G). The Grx1 system exhibited much lower decysteinylation activity than the Trx1 and TRP14 systems (Appendix Fig. S3F).

## TRP14 knockout protects the pancreas from increased protein cysteinylation and γ-glutamylcysteinylation upon acute pancreatitis in mice

Next, we assessed the possible impact of TRP14 knockout (KO) in mice using the acute pancreatitis (AP) model that is known to trigger a type of

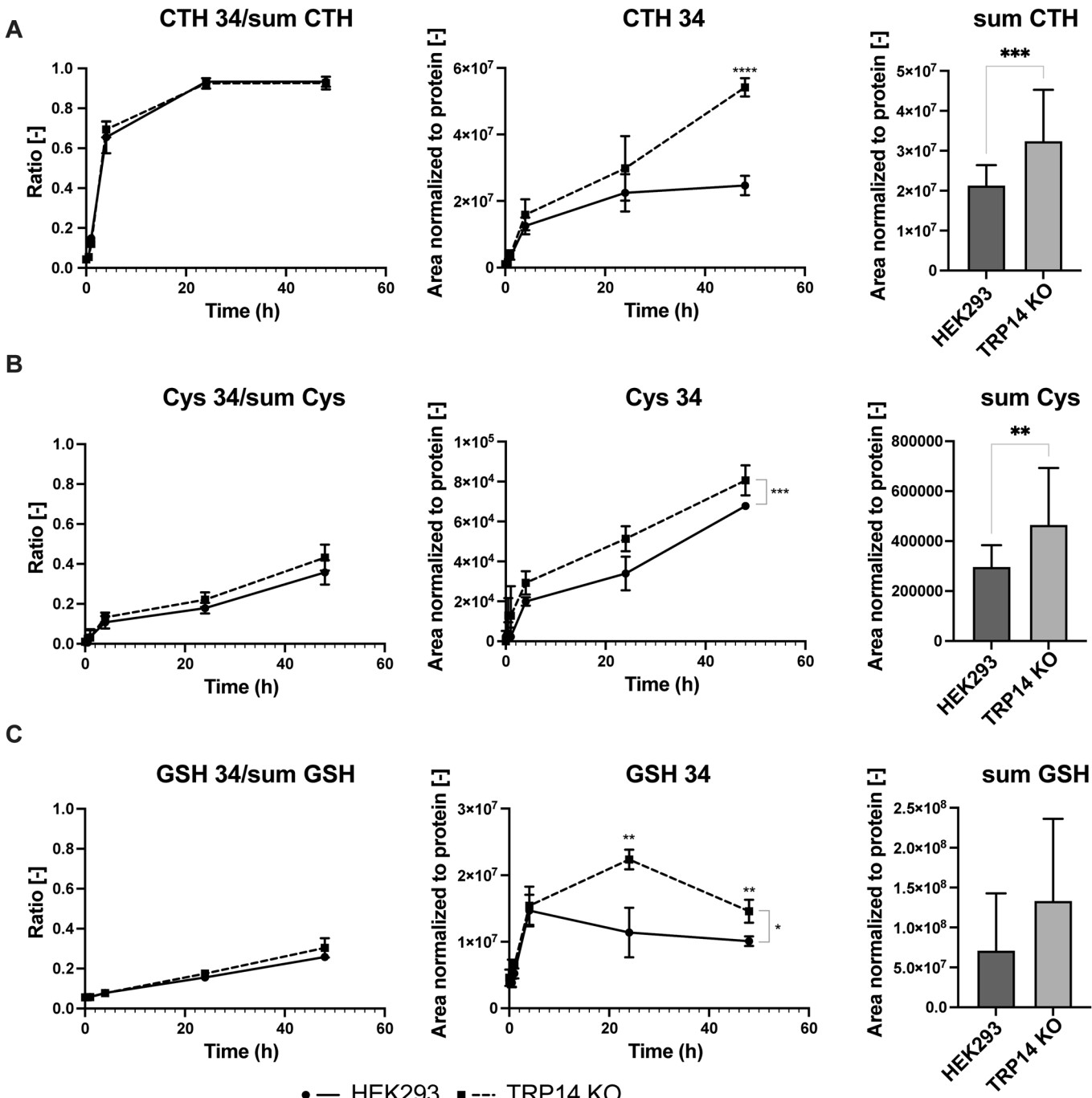

**Figure 2. Analysis of the flux through the transsulfuration pathway in HEK293 and TRP14 KO cells using heavy sulfur-containing methionine.**

Ratio of total (left), steady-state levels of heavy (middle) and steady-state levels of total (right) cystathionine (CTH) (**A**), cysteine (**B**), and glutathione (**C**) under cystine deprivation conditions (4 µM cystine in the culture medium) in HEK293 (solid lines) and TRP14 KO (dashed lines) cells. $n = 4$ (left and center panels (**A–C**), $n = 20$ (right panel (**A–C**). In the flux analysis, outliers according to Grubbs' test were excluded. Data information: Two-way ANOVA and Tukey's test for multiple comparisons: *$P < 0.05$; **$P < 0.01$; ***$P < 0.001$; ****$P < 0.0001$. The indicated "$n$" corresponds to the number of biological replicates. Data plotted corresponds to the mean (columns) ± standard deviation (error bars). Source data are available online for this figure.

oxidative stress characterized by increased protein cysteinylation (Moreno et al, 2014). TRP14 protein levels did not change significantly in the pancreas in AP of wild-type control mice (see Appendix Fig. S3G). At basal conditions, we found no alteration in the level of protein cysteinylation in the pancreas of TRP14 KO mice as compared to controls. Surprisingly, however, upon induction of AP protein cysteinylation levels in the pancreas increased only in control animals, but not in those lacking TRP14 (Fig. 7A). Similar effects were seen for protein γ-glutamylcysteinylation, while no changes in protein glutathionylation were observed in any of the four experimental groups (Fig. 7A).

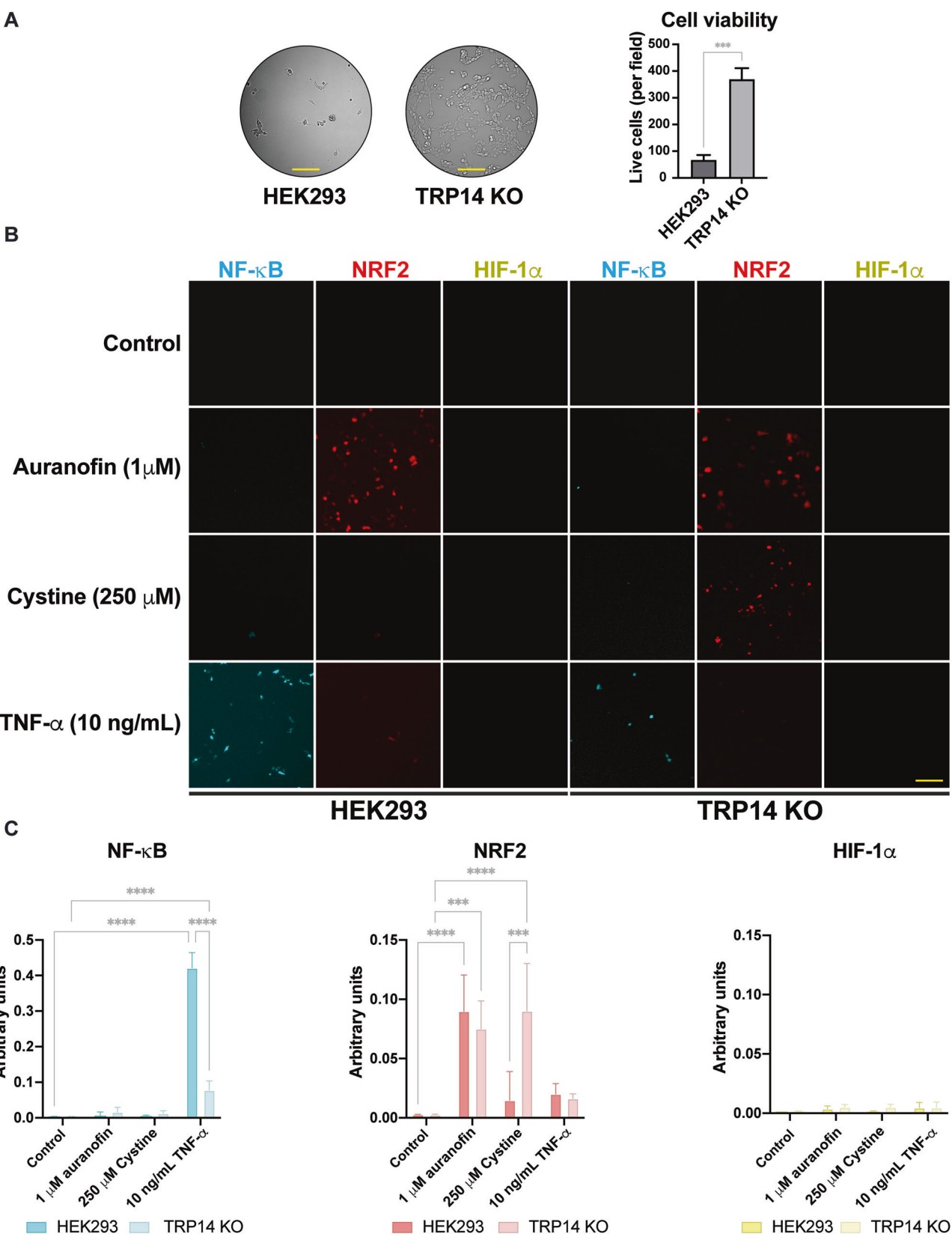

**Figure 3. TRP14 knockout leads to increased viability and Nrf2 activation upon exposure of HEK293 cells to super-physiological cystine concentrations.**

(A) Cell viability of HEK293 and TRP14 knockout cells incubated with 500 µM cystine for 24 h. (B) Representative images of NF-κB (cyan), NRF2 (red), and HIF-1α (yellow) activation in pTRAF-transfected HEK293 and TRP14 KO cells treated with 250 µM cystine for 24 h, with 1 µM auranofin and 10 ng/mL TNF-α used as positive controls. (C) Histograms corresponding to the quantification of the pTRAF signals. $n = 3$. Data information: Two-way ANOVA and Tukey's test for multiple comparisons: ***$P < 0.001$; ****$P < 0.0001$. The indicated "$n$" corresponds to the number of biological replicates. Data plotted corresponds to the mean (columns) ± standard deviation (error bars). Scale bar: 500 µm. Source data are available online for this figure.

We had initially expected that TRP14 KO would lead to increased protein cysteinylation levels in the pancreas, due to a lack of TRP14-mediated reduction of cystine and of cysteinylated proteins. Attempting to explain the unexpected effects of TRP14 knockout and aiming to assess alternative sources of reducing power and cysteine in the absence of TRP14, we next evaluated GSH levels, transsulfuration pathway proteins, and the Nrf2 status in these animals.

## GSH and cysteine levels were not depleted, and cystine levels rise in the pancreas from TRP14 knockout mice upon acute pancreatitis

The levels of GSH and oxidized glutathione-disulfide (GSSG) under basal conditions did not change in the pancreas upon TRP14 knockout. After AP induction in WT mice, pancreatic GSH levels decreased, while pancreatic GSH levels were maintained in TRP14 KO mice with AP (Fig. 7B). The GSSG to GSH ratio showed no significant changes between WT and TRP14 KO mice under basal conditions, but upon AP this ratio increased in WT as compared to TRP14 KO (Fig. 7B).

Pancreatic cysteine levels markedly dropped in wild-type mice with AP, but were maintained in TRP14 KO mice; the cystine levels also rose dramatically upon AP in TRP14 KO, although the cystine-to-cysteine ratio did not change significantly as compared to control animals with AP (Fig. 7C). No differences were observed in the levels of γ-glutamylcysteine or its oxidized form γ-glutamylcystine in the pancreas from WT and TRP14 KO mice under basal conditions. However, upon AP γ-glutamylcysteine levels were lower in WT mice than in TRP14 KO mice, thus increasing the γ-glutamylcystine to γ-glutamylcysteine ratio (Fig. 7D). γ-glutamylcystine levels were increased in TRP14 KO mice with AP compared to basal conditions, but the γ-glutamylcystine to γ-glutamylcysteine ratio remained unchanged.

Together, these results showed that cystine levels increased upon AP in TRP14 KO mice, which were in line with our initial expectations. Furthermore, the maintenance of GSH steady-state levels infered a good availability of cysteine, suggesting that an alternative route for intracellular cysteine supply might have been upregulated in the TRP14 KO mice with AP. We therefore next evaluated the key components of the transsulfuration pathway in these mice, as has been verified as an alternative source of cysteine in the absence of cystine-disulfide reduction activity (Eriksson et al, 2015) and an upregulation of transsulfuration would also agree with the HEK293 cell and *C. elegans* results described above.

## Both the transsulfuration pathway and Nrf2 activity are upregulated upon acute pancreatitis in the pancreas of TRP14 KO mice

We found that methionine levels were lower in the pancreas of TRP14 KO mice already under basal conditions, which further

dropped upon AP (Fig. 8A). Methionine levels were also lower in the pancreas from wild-type mice with pancreatitis as compared to basal conditions (Fig. 8A). No differences were found in homocysteine levels among the experimental groups (Fig. 8B), while cystathionine levels were markedly increased upon AP induction in TRP14 KO mice (Fig. 8C). AP also led to somewhat increased levels of cystathionine β-synthase (CBS) in both control as well as TRP14 KO animals, yet this was more pronounced in the KOs (Fig. 8D).

These findings showed that the transsulfuration pathway becomes activated in the pancreas upon AP, and much more so in mice lacking TRP14. We next evaluated the expression of the principal players of the Trx system, Trx1 and TrxR1, as well as that of additional Nrf2 target genes, considering that activated Nrf2 could promote both transsulfuration and increased GSH synthesis (Lu, 2009; Muri and Kopf, 2021).

Although no significant differences in the mRNA levels of *Txn1* and *Txnrd1* were observed in the pancreas from WT and TRP14 KO mice under basal conditions, these transcripts increased upon AP in both groups, and more markedly so in TRP14 KO mice (Fig. 8E,G). Also, the protein levels of Trx1 were much higher in the pancreas from TRP14 KO mice with AP than in WT mice with AP (Fig. 8F). Even under basal conditions the protein levels of Trx1 were higher in TRP14 KO mice than in WT mice, similar to our prior findings in HEK293 cells.

Analyzing the transcripts of sentinel Nrf2 target genes, no significant changes in the mRNA levels of NADPH-quinine oxidase-1, glutamate-cysteine ligase catalytic subunit, or heme-oxygenase-1 (*Nqo1*, *Gclc*, and *Ho1*, respectively) were observed in pancreas from WT or TRP14 KO mice under basal conditions (Fig. 8H–J). In WT mice with AP, *Gclc* and *Ho1* mRNA levels were higher than under basal conditions, but this increase was much less than that seen in TRP14 KO mice (Fig. 8H–J). Also, Nrf2 protein levels seemed higher in pancreas from TRP14 KO mice with AP, in comparison with basal conditions, furthermore displaying more nuclear translocation indicative of active Nrf2 (Fig. 8K,L).

## TRP14 knockout in mice leads to markedly lower pancreatic inflammation in acute pancreatitis; protection is lost upon blockage of transsulfuration

No differences in the histological analysis were observed between the pancreas from WT or TRP14 KO mice under basal conditions. AP typically triggers severe inflammatory responses and, accordingly, pancreas from WT mice with AP exhibited both tissue edema and inflammatory infiltrate. However, rather strikingly, TRP14 KO mice with AP showed only slight edema and almost no inflammatory infiltrate (Fig. 9A–C). An independent quantifiable indicator of inflammatory infiltrate, myeloperoxidase (MPO) activity (Schierwagen et al, 1990; Pérez et al, 2019), was also higher

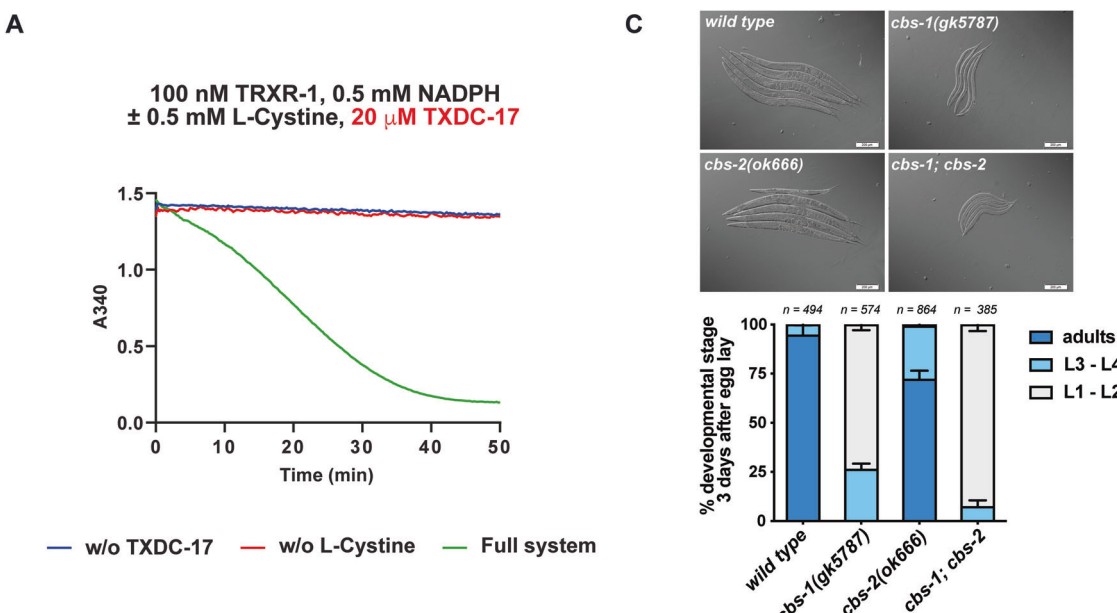

**A** 

**100 nM TRXR-1, 0.5 mM NADPH ± 0.5 mM L-Cystine, 20 μM TXDC-17**

— w/o TXDC-17    — w/o L-Cystine    — Full system

**C**

*wild type*    *cbs-1(gk5787)*

*cbs-2(ok666)*    *cbs-1; cbs-2*

■ adults  ■ L3 - L4  □ L1 - L2

**B**

| *C. elegans* strains | [a]Day at adulthood |
|---|---|
| Wild type | 3 |
| *txdc-17(syb4767)* | 3 |
| *cbs-1(gk5787)* | 6 to 7 |
| *cbs-2(ok666)* | 3 |
| *cth-1(ok3319)* | 3 |
| *cth-2(mg599)* | 3 |
| *cbs-1(gk5787); cbs-2(ok666)* | 7 |
| *txdc-17(syb4767); cth-1(ok3319)* | 3 |
| *txdc-17(syb4767); cht-2(mg599)* | 3 |
| *cth-1(ok3319); cth-2(mg599)* | 3 |
| *txdc-17(syb4767); cth-1(ok3319); cth-2(mg599)* | 3 |
| *rmIs133 [Q40::yfp]* | 4 |
| *txdc-17(syb4767); rmIs133 [Q40::yfp]* | 4 |
| *cth-1(ok3319); rmIs133 [Q40::yfp]* | 4 |
| *cth-2(mg599); rmIs133 [Q40::yfp]* | 4 |
| *txdc-17(syb4767); cth-1(ok3319); rmIs133 [Q40::yfp]* | 4 |
| *txdc-17(syb4767); cth-2(mg599); rmIs133 [Q40::yfp]* | **NOT VIABLE** |
| *unc-52(e669su250)* | 3 |
| *txdc-17(syb4767); cth-1(ok3319); unc-52(e669su250)* | 3 |
| *cth-2(mg599); unc-52(e669su250)* | 3 |
| *txdc-17(syb4767); cth-2(mg599); unc-52(e669su250)* | **NOT VIABLE** |

[a]The time at which egg-lay ended was set as 0

**Figure 4. Importance of TRP14 and the transsulfuration pathway in *C. elegans*.**

(**A**) Cystine reduction by the *C. elegans* TXRXR-1/TRP14 coupled enzyme system (green). (**B**) Table showing growth and viability of *C. elegans* mutants. (**C**) Quantification of *cbs-1* and *cbs-2* mutants growth. The graph shows the developmental stages of *cbs-1* and *cbs-2* mutants after 3 days incubation at 20 °C from a synchronized egg-lay. The micrographs are representative images of each genotype at day 3 after synchronized egg-lay. $n = 248$–$1157$ (**B**), $n = 385$–$564$ (**C**). Data information: The indicated "*n*" corresponds to the number of worms used. Data plotted corresponds to the mean (columns) ± standard deviation (error bars). Source data are available online for this figure.

in both WT and TRP14 KO mice with AP than under basal conditions but, again, TRP14 KO mice with AP displayed lower MPO activity than controls (Fig. 9D). Remarkably, blockage the transsulfuration pathway by inhibition of cystathionine β-synthase and cystathionine γ-lyase with aminooxyacetic acid (AOAA) led to

increased pancreatic edema and increased inflammatory infiltrate in TRP14 KO mice with pancreatitis when compared to TRP14 KO mice with pancreatitis but without AOAA treatment (Fig. 9E–G), which was confirmed by immunohistochemistry using the anti-CD11b/Integrin alpha antibody as a marker for neutrophil

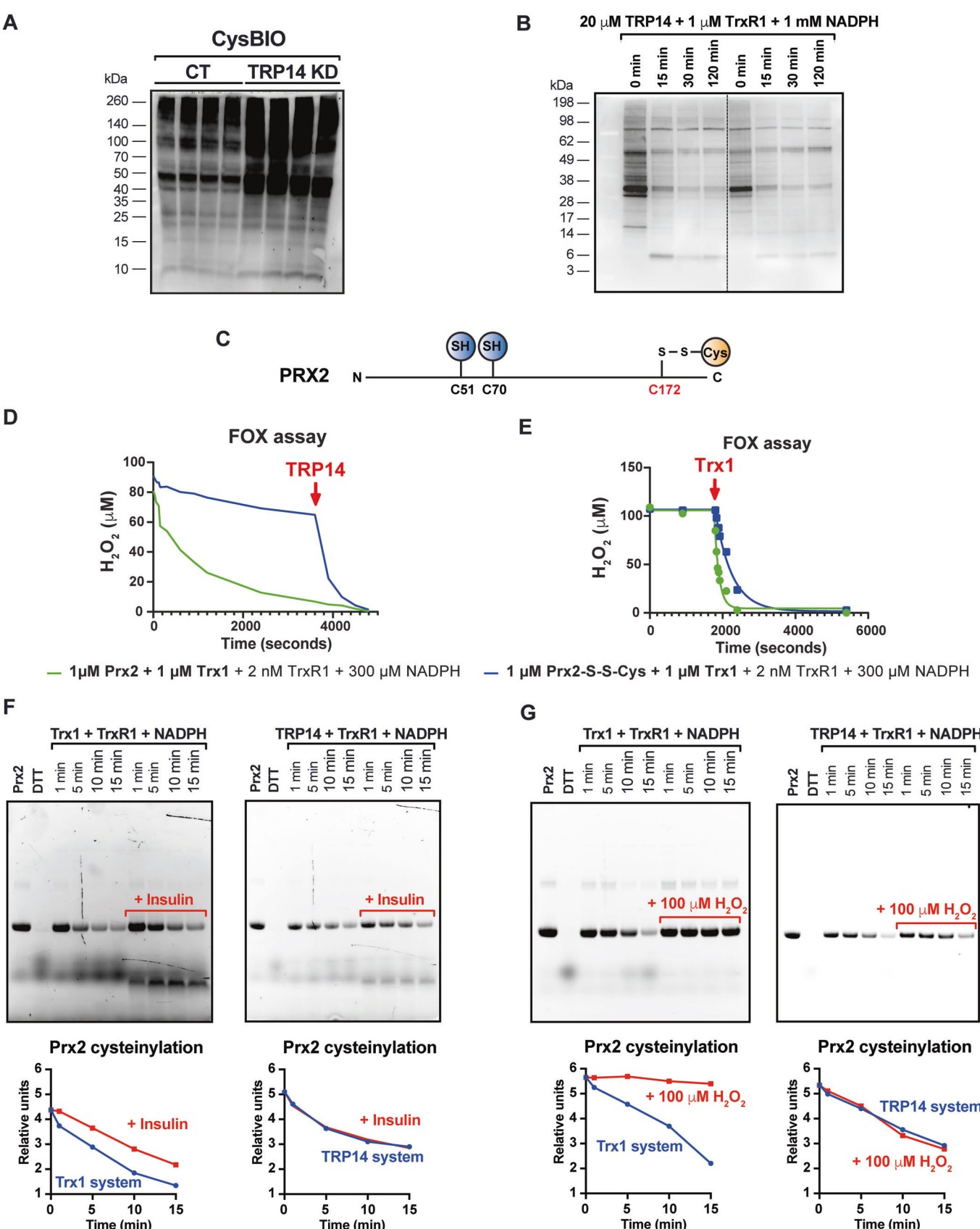

**Figure 5.  Protein cysteinylation in TRP14 knockdown HEK cells and decysteinylating activity of the TRP14 and Trx1 enzymatic system.**

(A) Representative western blot image showing increased protein cysteinylation levels in TRP14 knockdown HEK293 (TRP14 KD) cells compared to control (parental HEK cells) when incubated with biotinylated cysteine (Cys-BIO). Loading control in Appendix Fig. S3A. (B) Representative western blot image showing the streptavidin-HRP signal after incubating proteins labeled with Cys-BIO from two different cell lysates with the TRP14 enzymatic system for 15–120 min. Loading control in Appendix Fig. S3B. (C) Schematic representation of cysteine residue modifications detected by mass spectrometry in recombinant human peroxiredoxin 2 (Prx2) incubated with BODIPY™ FL L-Cystine. (D) Representative Fox assay to reduce $H_2O_2$ with 1 µM cysteinylated Prx2 (Prx2-S-S-Cys) (blue line) in the presence of TRP14 enzymatic system, and upon 10 µM TRP14 addition (red arrow); 1 µM Prx2 was used as a control (green line). (E) Representative Fox assay with 1 µM Prx2 (green line) and 1 µM Prx2-S-S-Cys (blue line) activity to reduce $H_2O_2$ in the presence of TRP14 enzymatic system; and after addition of 10 µM Trx1 (red arrow). (F) Effect of the presence of typical Trx1 substrates on the rate of the decysteinylation reaction using cysteinylated Prx2 as a substrate: the example of insulin. Representative fluorescence image showing that the addition of insulin into the reaction medium resulted in a decreased rate of decysteinylation by the Trx1 system when compared to cysteinylated Prx2 alone. However, insulin addition did not affect the decysteinylation rate of TRP14 (right). The graphs show the densitometric analysis corresponding to Prx2 cysteinylation levels. Loading control in Appendix Fig. S3E. (G) Effect of hydrogen peroxide on the decysteinylation rate by the Trx1 and TRP14 systems. Representative fluorescence images showing that the addition of 100 µM hydrogen peroxide impaired the decysteinylation reaction by the Trx1 system (10 µM Trx1, 10 nM TrxR1, and 1 mM NADPH), whereas it did not affect the rate of the decysteinylation reaction by the TRP14 system (20 µM TRP14, 50 nM TrxR1, and 1 mM NADPH). The graphs correspond to the densitometric analysis of Prx2 cysteinylation levels. Loading control in Appendix Fig. S3E. $n = 4$. Source data are available online for this figure.

infiltration (Appendix Fig. S4). These findings suggest that the milder form of the disease observed in TRP14 KO mice is, at least in part, due to an enhanced transsulfuration pathway activity.

## Effects of TRP14 knockout on cysteinylated proteins and global proteome of the pancreas in acute pancreatitis

In a proteomic analysis of the pancreatic tissue, we searched for cysteinylated proteins. We were able to confirm cysteinylation of peroxiredoxins 1, 2 and 4, and translation elongation factor 2 (EF-2, Appendix Tables S1–S4), but the quantitatively low levels of protein cysteinylation (only around 0.2% of the assigned spectra corresponded to cysteinylated peptides) hampered differential analysis of these modified peptides between experimental groups. Interestingly, however, three proteins (ATPase subunit b, ribophorin-1, and tyrosyl-tRNA synthetase) were found in cysteinylated form only in TRP14 KO mice under basal conditions, whereas voltage-dependent anion-selective channel protein 2, p100 co-activator, L-lactate dehydrogenase, and carboxypeptidase A1 were cysteinylated in both WT and TRP14 KO mice under basal conditions (Appendix Tables S1–S4). The overall profile of cysteinylated proteins however changed dramatically in mice with AP (Appendix Tables S1–S4).

Evaluating the complete proteome in the pancreatic tissue samples from the different animal groups, TRP14 KO mice showed a major alteration of the global proteome, with a large number of proteins showing significantly altered levels both under basal conditions and in AP, also confirming the upregulation of Nrf2 targets in TRP14 KO mice with AP, including Gclc and glutathione S-transferase P1 (Gstp, Appendix Figs. S5–S9). Many of the altered proteins were related specifically to redox pathways, mitochondrial function, and protein synthesis (see "Discussion").

## Discussion

Many Trx-fold proteins are likely to be able to reduce cystine in vitro, but since no dedicated cystine reductase has been identified in mammalian cells, it has remained unclear how cystine in the cytosol is reduced to cysteine; here we found that TRP14 is the rate-limiting enzyme for this function. With TRP14 being the major cytosolic reductase of cystine in human, C. elegans, and mouse cells, and considering that the transsulfuration pathway is the only reported alternative source of intracellular cysteine, these two cysteine sources

must be functionally coordinated and linked. Consistent with this, deletion of TRP14 has little impact on cell physiology as long as the transsulfuration pathway can provide cells with sufficient cysteine; however, TRP14 became essential in C. elegans under conditions of proteotoxic stress when transsulfuration was absent. Surprisingly, TRP14 knockout mice were protected from tissue damage in acute pancreatitis, with our findings suggesting that this was due to an exaggerated compensatory Nrf2 activation and upregulation of the transsulfuration pathway (Liu et al, 2020). Together, these results suggest that the transsulfuration pathway and TRP14 activity provide two alternative, complementary and important sources of cysteine for metazoan cells.

Here we also studied TRP14-catalyzed protein decysteinylation reactions. These seemed to be non-redundant with regards to other reductive pathways, as illustrated by the overall increased levels of cysteinylated proteins upon TRP14 knockdown in cells, or its knockout in mice with pancreatitis. The hereby identified cysteinylated proteins also suggest that protein cysteinylation does not necessarily occur at random, but may involve preferred target proteins, such as Prxs and EF2. Increased protein cysteinylation may have pathophysiological importance and has been linked to several different pathologies such as Alzheimer's disease (Poulsen et al, 2014; Costa et al, 2019), chronic kidney disease (Regazzoni et al, 2013; Ostrakhovitch and Tabibzadeh, 2015), liver cirrhosis (Domenicali et al, 2014), cardiovascular disease (Belcastro et al, 2017; Wakabayashi et al, 2017), and rheumatoid arthritis (Seward et al, 2011).

A functional redundancy was proposed between the Trx1 and GSH systems (Arnér, 2009; Eriksson et al, 2015) to allow survival upon failure of one of these systems as TrxR1-null mouse livers rely on GSH to provide electrons through glutaredoxin to ribonucleotide reductase for DNA synthesis (Holmgren, 1989, 19; Prigge et al, 2012), and either the Trx system alone when the GSH system is lacking, or the GSH system alone when the TrxR1/Trx1 is lacking, can support cell survival (Prigge et al, 2017). Here we found that TRP14 exhibits decysteinylating activity on many cellular proteins, and that it can regulate the activity of Prx2 through control of its cysteinylation state, although it cannot directly support Prx2 catalysis, in contrast to Trx1. Importantly, there is some redundancy in decysteinylation activity by several players in the Trx/TRP14 and GSH systems, which may explain the rather low level of protein cysteinylation found in TRP14 knockout mice under basal conditions, as well as in acute pancreatitis when considering a compensatory upregulation in the thioredoxin and GSH system.

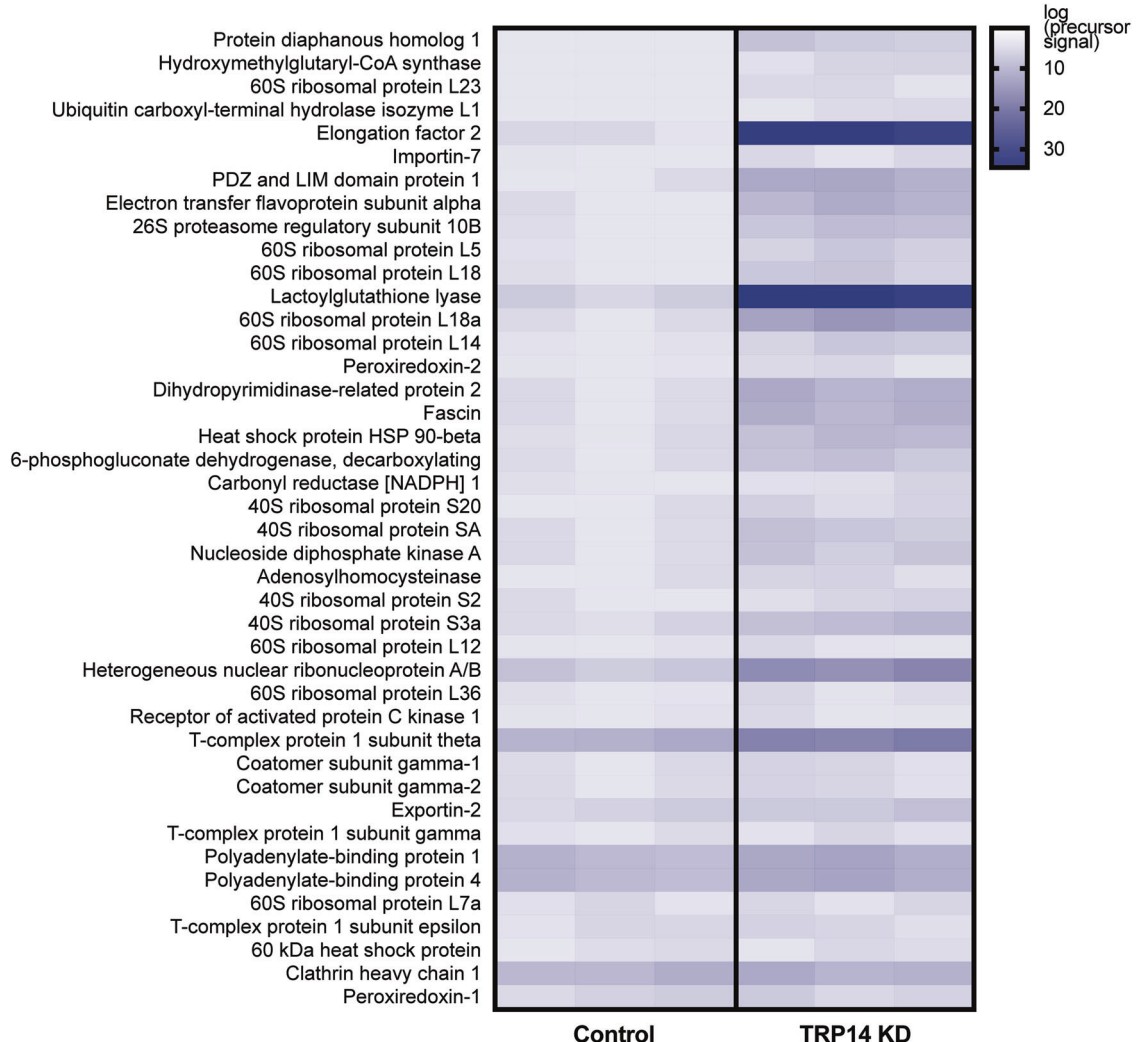

**Figure 6. Proteomic analysis of cysteinylated proteins in TRP14 knockdown cells.**

The proteomic analysis consisted of three different experiments in which wild-type cells (Control) and TRP14 knockdown cells (TRP14 KD) were incubated with biotinylated cysteine and then enriched for cysteinylated proteins using a streptavidin column. As a control for the technique, cells incubated with cysteine (no label) were processed in parallel to compensate for any nonspecific binding to the column. Once the proteomic data were retrieved, we worked with those proteins that were identified in both wild-type and TRP14 knockdown cells in the three independent experiments but not in the unlabeled samples. The cysteinylated peptides were identified, and the signal intensity of each detected precursor peptide was compared between groups. After performing a $t$ test, we identified 42 proteins that were significantly more cysteinylated in TRP14 KD cells than in WT cells ($P < 0.05$). These are the proteins shown in the heatmap, which shows the logarithm of the intensity of the precursor signal for each of the proteins (the darker the blue, the higher the signal). This signal is normalized to the sum of the intensities of all peptides detected in the sample. $n = 3$. Data information: T test where $P < 0.05$. The indicated "n" corresponds to the number of biological replicates. Data plotted corresponds to the log of the intensity of the precursor peptide signal (shade of blue) for each experiment (cells).

In conclusion, our study has identified TRP14 as the rate-limiting enzyme for intracellular cystine reduction, thereby supplying cells with cysteine in a pathway complementary to that using methionine through transsulfuration (Fig. 10). We have also shown that, in addition to its described roles in protein depersulfidation (Dóka et al, 2020), TRP14 has protein decysteinylation activity, with specific proteins being amenable to redox control through cysteinylation, as here shown with peroxiredoxin 2. Furthermore, TRP14 is a dispensable protein under basal conditions in human HEK293 cells, *C. elegans* as well as in mice. However, its knockout in *C. elegans*

together with deletion of transsulfuration becomes lethal under proteotoxic stress, while in mice the knockout of TRP14 confers *resistance* against tissue damage in acute pancreatitis through an increased activation of Nrf2 and the transsulfuration pathway under conditions of oxidative stress. Our findings collectively show that TRP14 constitutes a key regulatory redox node associated with the reduction of cystine into cysteine, and control of protein cysteinylation levels, clearly acting in concert with the transsulfuration pathway, and thus profoundly affecting cysteine homeostasis and the cellular redox proteome.

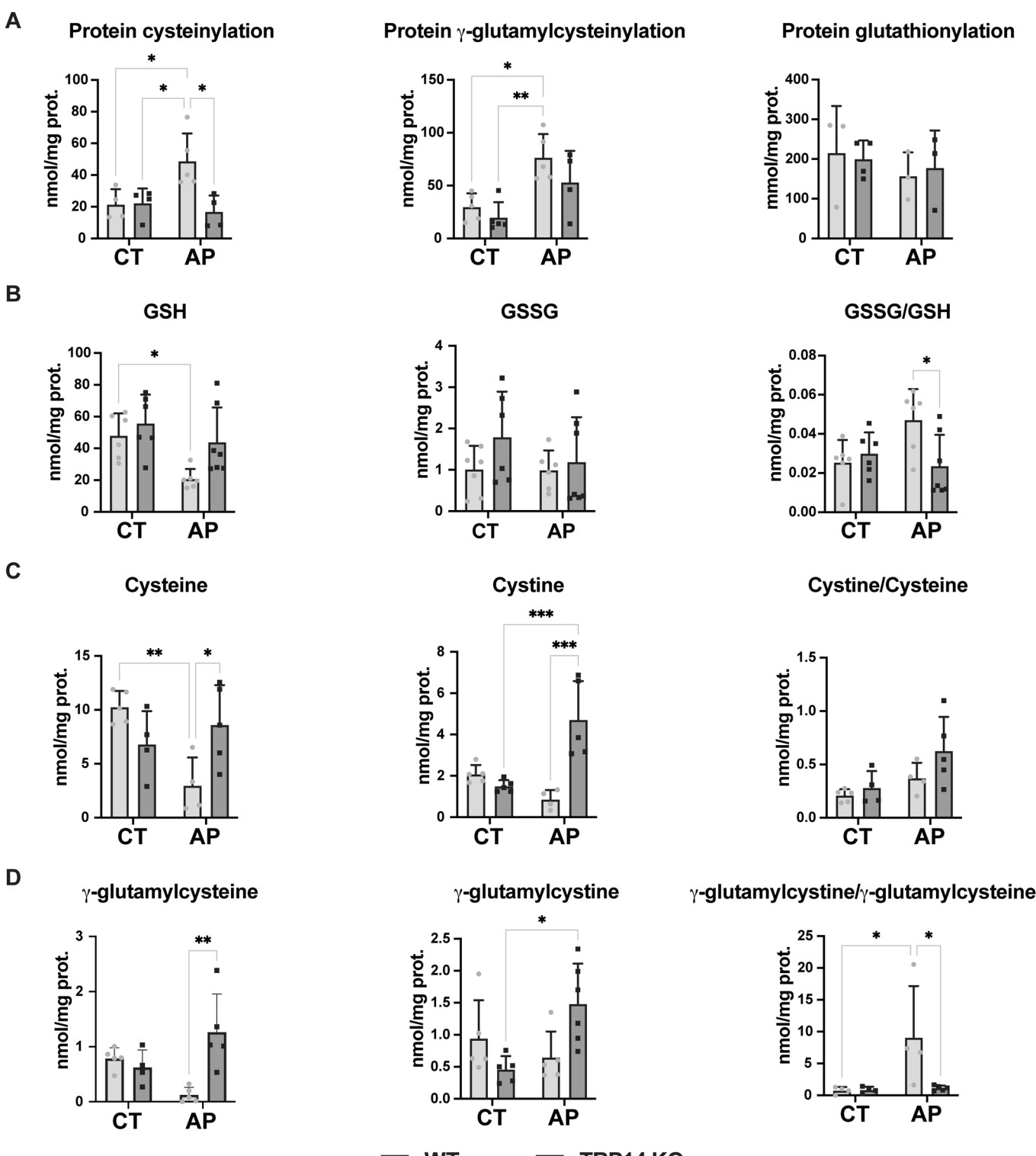

**Figure 7. Protein cysteinylation, and GSSG/GSH, cystine/cysteine, and γ-glutamylcystine/γ-glutamylcysteine redox pairs in pancreas from wild-type and TRP14 knockout mice.**

(A) Levels of protein cysteinylation, protein gamma-glutamylcysteinylation, and protein glutationylation ($n = 5$); (B) GSH and GSSG levels, and GSSG/GSH ratio ($n = 7$; outliers according to Grubbs' test were excluded); (C) cysteine and cystine levels, and cystine/cysteine ratio ($n = 5$); (D) γ-glutamylcysteine and γ-glutamylcystine levels, and γ-glutamylcystine/γ-glutamylcysteine ratio ($n = 5$), in pancreas from wild-type and TRP14 knockout (TRP14 KO) sham mice (control, CT) and upon acute pancreatitis (AP). *$P < 0.05$ vs. WT control; **$P < 0.01$ vs. WT control; #$P < 0.05$ vs. KO control; ##$P < 0.01$ vs. KO control; §$P < 0.05$ vs. WT pancreatitis; §§$P < 0.005$ vs. WT pancreatitis. Data information: Two-way ANOVA and Tukey's test for multiple comparisons: *$P < 0.05$; **$P < 0.01$; ***$P < 0.001$. The indicated "$n$" corresponds to the number of animals used. Data plotted corresponds to the mean (columns) ± standard deviation (error bars). Source data are available online for this figure.

# Methods

## Cloning, expression, and purification of human TRP14, Trx1, TrxR1, and Prx2, and *C. elegans* TRXR-1 and TRP14

Human TRP14, Trx1, TrxR1, and Prx2 were recombinantly expressed and purified as previously described (Pader et al, 2014; Dóka et al, 2016, 2020; Cheng and Arnér, 2017, 2018). The TRXR-1 selenoprotein of *C. elegans* and the corresponding TRP14 protein were expressed and purified with the same methodologies as the human orthologues, with the amino acid sequences given in Appendix Table S6.

## Cell culture and propagation of HEK293 cells

HEK293 cells (CRL-1573, ATCC, Virginia, USA) were cultured in Eagle's minimum essential medium (EMEM) (Gibco, MA, USA), supplemented with 10% fetal bovine serum (FBS) (Gibco, MA, USA), 100 U/mL penicillin (Gibco, MA, USA), 100 mg/mL streptomycin (Gibco, MA, USA), and 100 nM sodium selenite (Sigma-Aldrich, MO, USA). The culture medium contained 0.03120 g/L of L-Cystine dihydrochloride and 0.01500 g/L of L-Methionine. The complete media formulation is available from the manufacturer's website. Cells were incubated at 37 °C in humidified air containing 5% $CO_2$ and were kept in the logarithmic growth phase. Cells were tested for mycoplasma contamination.

## Generation of TRP14 knockout in HEK293 cells

TRP14 knockout cells were obtained from HEK293 using CRISPR/Cas9 technology, as constructed and performed at the Karolinska Genome Engineering (KGE) facility at Karolinska Institutet. In short, three different guide RNAs targeting the TXNDC17 locus were designed: CGGGCCGTGGAACAGCACAA-TGG, CCGGTG GAACTCCTCGAAGC-CGG and CGTGCCAATGGCCCGCTA TG-AGG. HEK293 cells were set in 24-well plates and transfected with 2.5 μg Cas9 (Alt-R® S.p. Cas9 Nuclease V3, IDT) and 1 μg guide RNAs (CRISPRevolution sgRNA EZ Kit, Synthego) pre-complexed in ribonucleoproteins. After transfection, single clones obtained after single-cell dilution selection in a 96-well plate were expanded and validated for TXNDC17 knockout using immunoblotting for TRP14.

## Reconstitution of TRP14 knockout cells

To reconstitute TRP14 KO cells, a human plasmid containing the *TXNDC17* sequence in a pCMV6-Entry vector from OrigGene (Maryland, USA) was used. HEK293 and TRP14 KO cells (10,000/well) were seeded in a 96-well plate and cultured overnight. Then, each well was transfected with 0.1 μg of the plasmid and 0.4 μL of TurboFect Transfection Reagent (ThermoFisher, MA, USA) in 20 μL of Opti-MEM™ (Gibco, MA, USA). The cells that underwent transfection were incubated with the transfection mixture for 48 h before conducting functional assays and western blotting.

## Cell viability measurement

Cells were seeded in 96-well plates (ThermoFisher, MA, USA) and let grow for 24 h. After incubation with 500 μM cystine for 24 h, cells were checked under the microscope, and the number of live cells was counted.

## Fluxomic analysis of low-molecular weight (LMW) metabolites

Cystine deprivation medium was made based on Gibco's Eagle's minimum essential medium (EMEM) formulation without cystine and methionine, and it was supplemented with 10% dialyzed fetal bovine serum (dFBS) (Sigma #F0392), 100 U/ml penicillin and 100 μg/ml streptomycin (Lonza #DE17-602E), 2 mM L-glutamine (Lonza #17-605E) and 100 nM sodium selenite (Sigma #S5261). In total, 4 or 200 μM cystine (Sigma #C7602) and 0.015 g/L [$^{34}$S]-methionine (Sigma #M9625) were added freshly before the experiment. Cells were seeded in 6 well plates 24 h before cystine deprivation and culture medium was changed to heavy sulfur [$^{34}$S]-methionine-containing deprivation medium 0, 0.5, 1, 4, 24 and 48 h before cell lysis.

Measurement is based on the method published by Akaike et al, (Akaike T et al, 2017) as described here: Cells were washed twice with cold HBSS and they were harvested in ice-cold 75% methanol solution containing 5 mM β-(4-hydroxyphenyl)ethyl iodoacetamide (HPE-IAM). After sonication, the alkylation was carried out at 37 °C for 20 min and followed by centrifugation (14,000 × $g$; 10 min; 4 °C). The supernatant was acidified with 10% formic acid and diluted two-fold with 0.1% $FA/H_2O$ before injection. Cell pellets were dissolved in 1% SDS/PBS, sonicated and protein content was measured using BCA assay.

Liquid chromatography-tandem mass spectrometry (LC-MS/MS) measurements were carried out on a Thermo Q-Exactive Focus Orbitrap mass spectrometer coupled to a Thermo Vanquish UHPLC (ultra-high-performance liquid chromatograph), and samples were analyzed with two different methods. MS/MS detection was carried out in positive ionization mode, higher-energy collisional dissociation (HCD) was used to detect $^{32}$S cysteine ($m/z$ 299 > 121), $^{34}$S cysteine ($m/z$ 301 > 121), $^{32}$S glutathione ($m/z$ 485 > 356), $^{34}$S glutathione ($m/z$ 487 > 358), cystine ($m/z$ 241 > 152), $^{32}$S cystathionine ($m/z$ 223 > 134) and $^{34}$S cystathionine ($m/z$ 225 > 136).

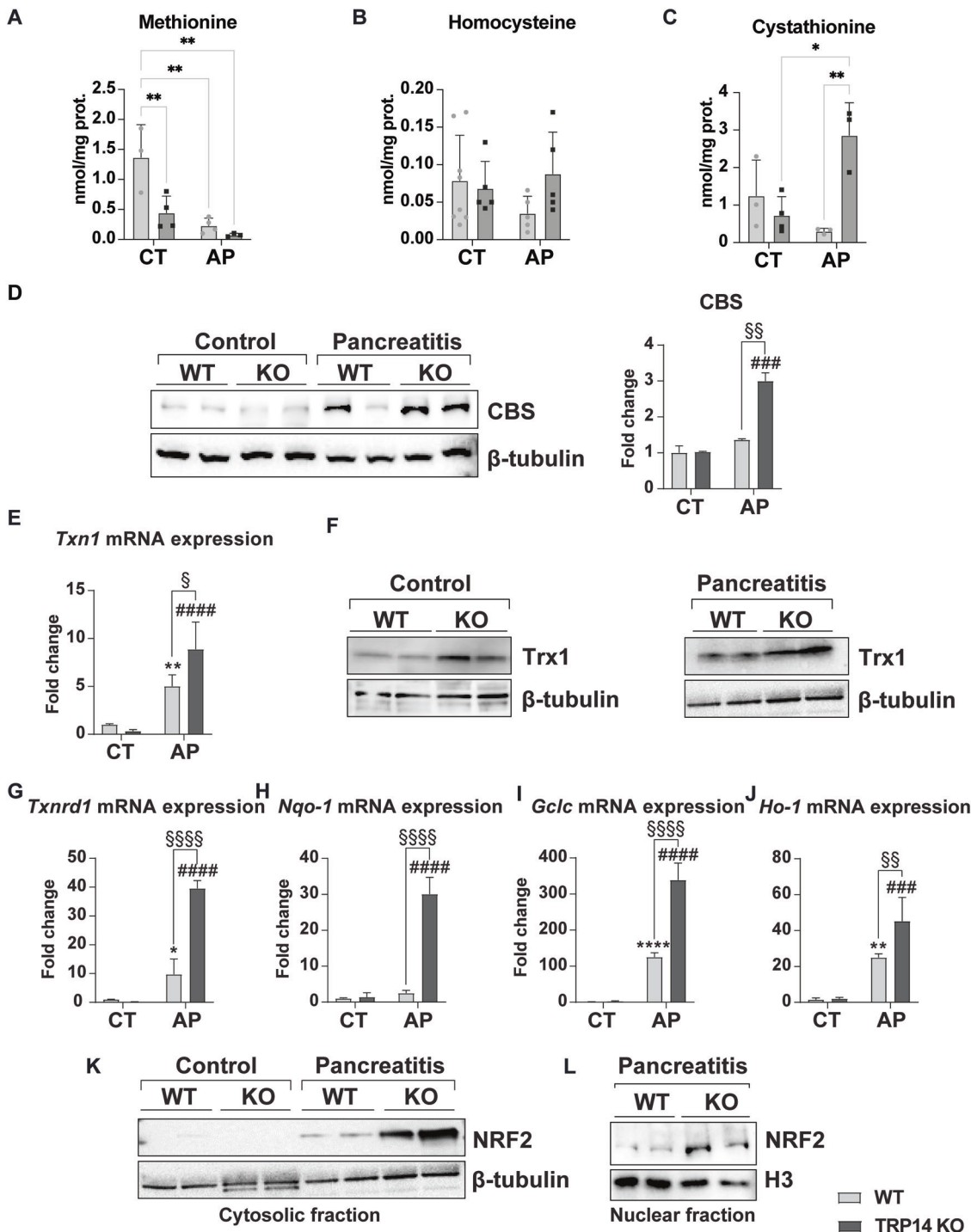

Measurement of derivatized analytes was carried out on a Phenomenex Kinetex C18 (50 × 2.1 mm, 2.6 μm) column with eluents 0.1% FA/H$_2$O (A) and 0.1% FA/MeOH (B). The initial 5% B was linearly increased first to 13% in 2 min, then to 95% in 4 min, held there for 0.5 min, then lowered back to 5% B in 0.1 min and held there for 3.4 min before the next injection. Flow rate was 0.5 ml/min, the column was thermostated at 40 °C.

Thermo Scientific Hypercarb (100 × 2.1 mm, 3 μm) column was used to detect cystine and cystathionine, with eluents of 0.5% FA/H$_2$O (A) and 0.5% IPA:ACN 1:1. (B). Initial 0% B was linearly increased to 3% in 1.5 min, then to 30% in 3.5 min, then to 100% in 1 min, kept for 2 min, then decreased to 0% B in 1 min, held for 8 min. The column temperature was 40 °C, and the flow rate was 0.2 ml/min.

◄ **Figure 8.  TRP14 deficiency triggers Nrf2 activation and upregulation of the transsulfuration pathway in the pancreas upon pancreatitis markedly diminishing the inflammatory response.**

(A) Methionine levels ($n = 4$). (B) Homocysteine levels ($n = 4$). (C) Cystathionine levels ($n = 4$). (D) Representative western blot for cystathionine β-synthase (CBS), and β-tubulin as loading control ($n = 4$). (E) Thioredoxin (*Txn1*) mRNA expression ($n = 5$). (F) Representative western blot for Trx1, and β-tubulin as loading control ($n = 4$). (G) Thioredoxin reductase 1 (*Txnrd1*) mRNA expression ($n = 6$). (H) NAD(P)H dehydrogenase quinone oxidoreductase 1 (*Nqo1*) mRNA expression ($n = 4$). (I) Glutamate-cysteine ligase catalytic subunit (*Gclc*) mRNA expression ($n = 4$). (J) Heme-oxygenase 1 (*Ho1*) mRNA expression ($n = 5$). (K) Representative western blot of NRF2 cytosolic levels in the pancreas ($n = 4$). (L) Representative western blot for nuclear NRF2 levels upon acute pancreatitis induction ($n = 4$). Data information: Two-way ANOVA and Tukey's test for multiple comparisons: (A–C) *$P < 0.05$; **$P < 0.01$; (D, E, G–J) *$P < 0.05$ vs. WT CT; **$P < 0.01$ vs. WT CT; ***$P < 0.0001$ vs. WT CT; ###$P < 0.001$ vs. TRP14 KO CT; ####$P < 0.0001$ vs. TRP14 KO CT; §$P < 0.05$; §§$P < 0.01$; §§§§$P < 0.0001$. The indicated "*n*" corresponds to the number of animals used. Data plotted corresponds to the mean (columns) ± standard deviation (error bars). Trx1 and NRF2 were analyzed on the same blots, and hence in (F, K) the tubulin loading control was the same for the pancreatitis samples. Source data are available online for this figure.

## pTRAF (plasmid for transcription factor reporter activation based upon fluorescence) assays

Briefly, wild-type and TRP14 knockout cells (10,000/well) were seeded in Corning® BioCoat® Collagen I-coated plates (Corning, NY, USA) suitable for microscopy and let to grow overnight. Then, cells were transfected with the plasmid as previously described by us (Johansson et al, 2017), and 24 h later, the medium was changed, and the treatments were added (1 μM auranofin, 20 ng/mL TNF-α, and 250 μM cystine). Imaging was performed using a Leica DMi8 fluorescence microscope (Leica, Wetzlar, Germany) 24 h after treatment addition, and the acquired images were analyzed using CellProfiler™ (Broad Institute, MA, USA).

## Reduction of BODIPY™ FL L-Cystine in cells

Wild-type and TRP14 knockout cells (10,000/well) were seeded in 96-well plates suitable for microscopy and let grow for 24 h under the previously detailed standard culture conditions. Then, both cell types were incubated with 1 nM BODIPY™ FL L-Cystine (Thermo-Fisher, MA, USA) for 30 min at 37 °C, and cystine reduction—observed here as an increase in green fluorescence within the cells—was assessed under a Leica DMi8 fluorescence microscope (Leica, Wetzlar, Germany).

The reduction capacity of BODIPY™ FL L-Cystine was also assessed under different experimental conditions using a TECAN Infinite® 200 PRO plate reader (TECAN, Switzerland) by measuring the fluorescence signal at one-minute intervals. The slope of the curves obtained was normalized by the protein concentration in each well. The detection wavelengths used were 505 nm excitation and 520 nm emission.

## Animals

### C. elegans strains
The standard methods used for culturing and maintenance of *C. elegans* were as previously described (Stiernagle, 2006). A list of all strains used and generated in this study is provided in Appendix Table S5. The *txdc-17* allele *syb4767* was generated at SunyBiotech (http://www.sunybiotech.com) by CRISPR-Cas9 editing. All VZ strains are 6× outcrossed with N2 wild-type, except the strain generated by CRISPR-Cas9 (*syb4767* allele) that was 2× outcrossed. Worm reagents and details on the protocols used for genotyping the different alleles reported in this work can be provided upon request.

### C. elegans development quantification
Synchronized worms were generated by allowing 10–15 gravid hermaphrodites to lay eggs during 2–3 h on *E. coli* OP50 seeded plates at 20 °C. Then parents were removed and development of the progeny was scored the day at which animals reached adulthood (fertilized embryos in the uterus) for Fig. 4B and at day 3 after egg-lay for Fig. 4C. Differential interphase contrast micrographs were acquired at 10x magnification in an Olympus BX61 microscope equipped with a DP72 digital camera coupled to CellSens Software for image acquisition and analysis. Photoshop CC 2018 and Adobe Illustrator software were used to produce the final micrographs.

## TRP14 knockout mice

C57BL/6 *Txndc17* wild-type mice encoding TRP14 (WT) as well as the *Txndc17* null mice (KO) were described previously (Dóka et al. 2020). Male WT and KO mice used in this study were fed a standard chow diet and water *ad libitum*, and were housed at a temperature of 22 ± 1 °C, 60% relative humidity, and constant 12 h/12 h light/darkness cycles. All animal care and procedures in this study were reviewed and approved by the corresponding oversight authority and were performed in compliance with the regulations of the European Parliament and of the Council of September 22, 2010 on the protection of animals used for scientific purposes (Directive 2010/63/EU), incorporated into the Spanish law by the Real Decreto 53/2013 (RD 53/2013) from February 1, 2013. Mouse procedures done at Montana State University (MSU) were performed under MSU Institutional Animal Care and Use Committee (MSU IACUC) protocols 2019-50-97, 2022-50-IA, or 2021-118-01. Mouse procedures done at the University of Veterinary Medicine Budapest were performed under Hungarian animal ethics protocols PE/EA/00744-6/2022 or PE/EA/00977-6/2023. Mouse procedures done at the animal housing facilities from the SCSIE located at the Faculty of Pharmacy from the Universitat de València with protocols 2020/VSC/PEA/0030 and 2023-VSC-PEA-0201 were approved by the Ethics Committee for Animal Welfare from the Universitat de València.

## Synthesis of biotinylated cysteine

To a stirred solution of (+)-Biotin N-hydroxysuccinimide ester (100 mg, 0.29 mmol) in 70% MeOH (5 mL), a solution of L-cysteine (70 mg, 0.58 mmol) in 10 mM $NaH_2PO_4$ (5 mL) was added. pH of the reaction was adjusted to 8 by adding an NaOH solution (1 M), then, the reaction was stirred for 2 h at room temperature.

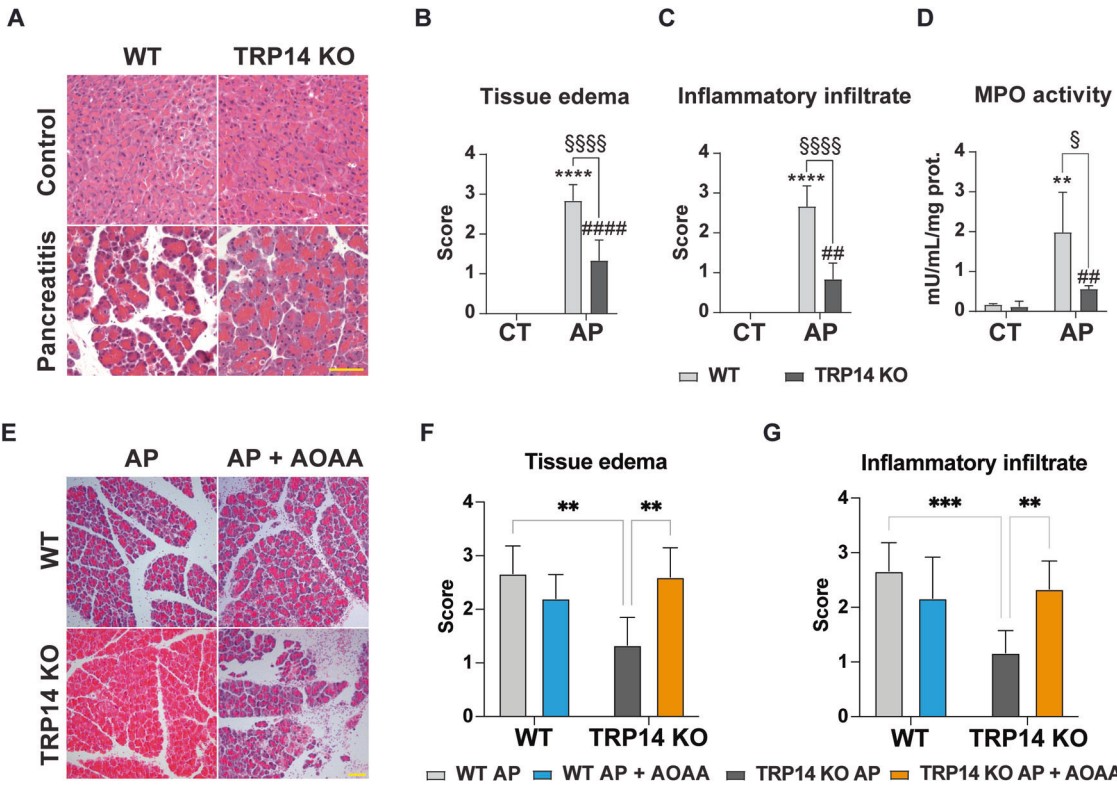

**Figure 9.  TRP14 knockout confers protection against acute pancreatitis in mice, and inhibition of the transsulfuration pathway abrogated the protective effect in TRP14 KO mice.**

(A) Representative hematoxylin–eosin staining; (B) tissue edema histological score; (C) inflammatory infiltrate histological score; and (D) Myeloperoxidase (MPO) activity in pancreas from wild-type and TRP14 knockout mice under basal conditions and in cerulein-induced acute pancreatitis. (E) Representative hematoxylin–eosin staining; (F) tissue edema histological score; and (G) inflammatory infiltrate histological score in wild-type and TRP14 knockout mice with pancreatitis with and without AOAA. $n = 3$–6. Data information: Two-way ANOVA and Tukey's test for multiple comparisons: (B–D) **$P < 0.01$ vs. WT CT; ****$P < 0.0001$ vs. WT CT; ##$P < 0.01$ vs. TRP14 KO CT; ####$P < 0.0001$ vs. TRP14 KO CT; §$P < 0.05$; §§§§$P < 0.0001$; (F, G) **$P < 0.01$; ***$P < 0.001$. The indicated "$n$" corresponds to the number of animals used. Data plotted corresponds to the mean (columns) ± standard deviation (error bars). Scale bar: 100 μm. Source data are available online for this figure.

Thereafter, the reaction was cooled down to 4 °C, and the mixture was adjusted to pH 2 by adding an HCl solution (1 M) and kept overnight at 4 °C. The resulting biotinyl-cysteine precipitate formed was filtered, washed with $H_2O$ and re-crystalized in 10 mM $NaH_2PO_4$ to yield a white precipitate (87 mg, 86% yield). HRMS (ES + ) mass calcd. for $C_{13}H_{21}N_3O_4S_2$ 347.0973; found $[(M + H) + ]$ 348.1040; found $[(2 M + H) + ]$ 695.2010. Elemental analysis calculated for $C_{13}H_{21}N_3O_4S_2O$: % C 44.94, % H 6.09, % N 12.09, % S 18.45; found: % C 44.44, % H 6.02, % N 11.95, % S 18.20.

## Identification and quantification of cysteinylated proteins in TRP14 knockdown cells

Cells ($4 \times 10^6$) were seeded in T-75 Falcon™ culture flasks (Corning, MA, USA) containing 10 mL of fresh medium (EMEM supplemented with 10% FBS, 100 U/mL penicillin, 100 mg/mL streptomycin, and 50 nM sodium selenite). Experiments were performed 24 h after seeding.

For the obtention of cysteinylated cell lysates, both control and TRP14 KD cell cultures were incubated with 250 μM biotinylated cysteine (Cys-BIO) for 1 h, the incubation medium was discarded,

cells were washed with ice-cold PBS and collected in 300 μL of lysis buffer containing 50 mM Tris-HCl (pH 7.5), 2 mM EDTA, 150 mM NaCl, 1% Triton, 10 mM N-ethylmaleimide (NEM), and a protease inhibitor cocktail (Sigma-Aldrich, MO, USA). Cells were lysed by three freezing-thawing cycles, and cell lysates were centrifuged at 3500 rpm for 10 min to collect supernatants for analysis. Pierce™ Monomeric Avidin Agarose Kit (Thermo Scientific, MA, USA) was used to purify cysteinylated proteins tagged with biotin according to the manufacturer's instructions.

The proteomic analysis of the enriched fractions was carried out at the Proteomics Section of the Central Service for Experimental Research (SCSIE) from the Universitat de València. Samples were dried in a rotary evaporator and 14 μg were loaded in a 1D PAGE gel to clean and concentrate the samples, which were later in-gel reduced with dithiothreitol (DTT) and S-alkylated with iodoacetamide (Sigma-Aldrich, Missouri, USA) as previously described (Shevchenko et al, 2006). Afterward, samples were digested with 100 ng sequencing-grade trypsin (Promega, Madison, USA). The digestion mixture was dried in a vacuum centrifuge and resuspended in 20 μL of 2% acetonitrile, and 0.1% trifluoroacetic acid (TFA). In total, 5 μL of sample were loaded onto a trap column (NanoLC Column, 3 μ C18-CL, 350 μm × 0.5 mm;

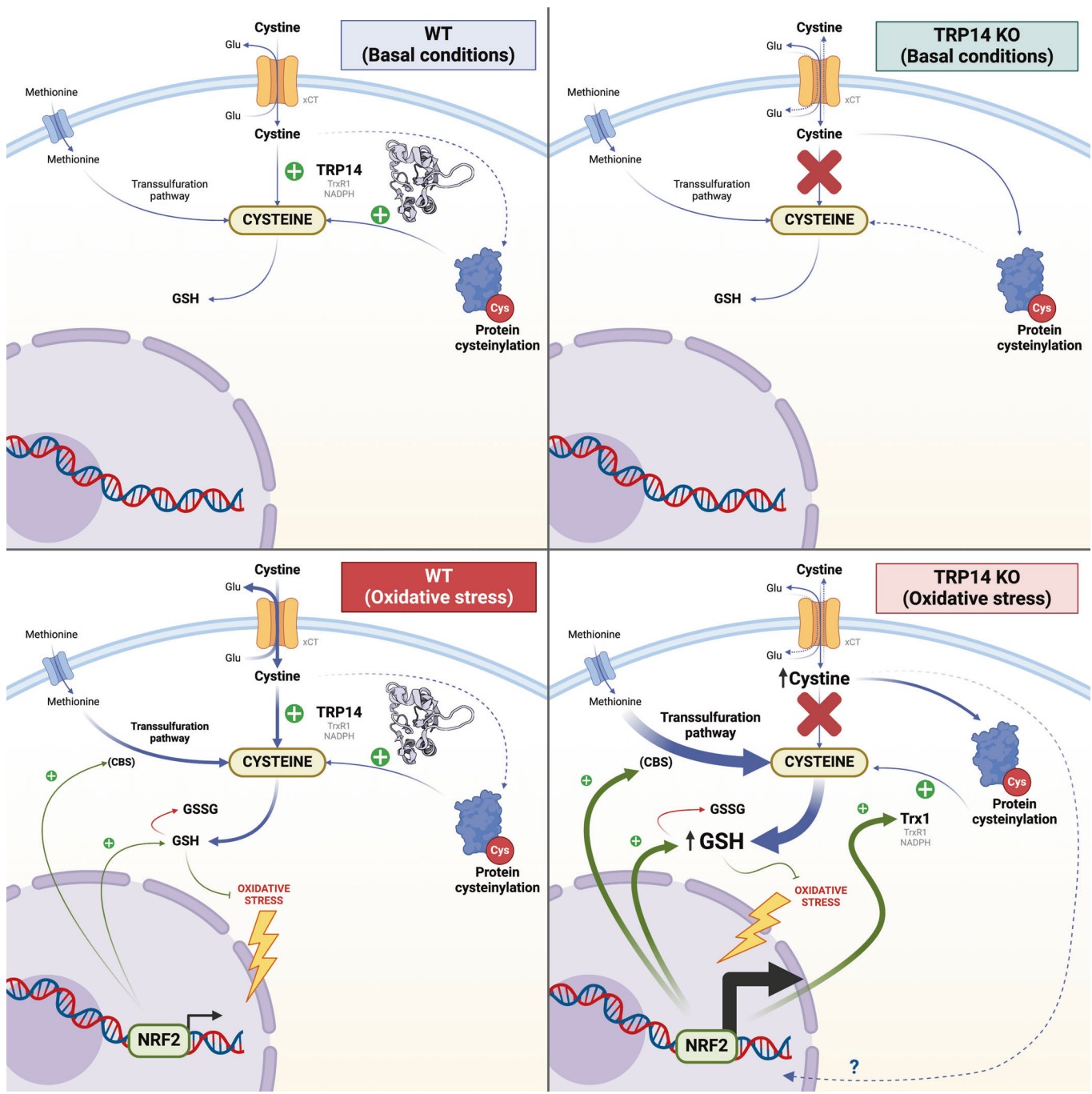

**Figure 10. Scheme of major findings in this study.**

Here we found that TRP14 reduces intracellular cystine to cysteine, as well as reduces protein cysteinylation motifs, being supported in cells by TrxR1 and NADPH. Cystine is taken up into cells through the xCT cystine/glutamate antiporter (orange). Cysteine can also be synthesized from methionine through the transsulfuration pathway, with methionine entering the cell via several transporter systems (blue). Cysteine is needed for GSH synthesis as well as protein synthesis, with both cystine and cysteine being able to cysteinylate proteins under oxidative stress conditions. Several of the involved enzymes and proteins, including CBS of the transsulfuration pathway, GSH synthesis enzymes, xCT itself, TrxR1, and many more enzymes, are target genes for Nrf2, which is typically activated upon oxidative stress. Here we found that knockout of TRP14 lacks major phenotypes under basal conditions due to the provision of cysteine through transsulfuration, although we could detect increased protein cysteinylation compared to wild-type cells (top two panels). Under conditions of oxidative stress, both wild-type and TRP14 knockout cells activate Nrf2 and thus the transsulfuration pathway as well as the Trx system, and here we show that the overall outcome of TRP14 knockout thus becomes context-dependent. In *C. elegans*, simultaneous knockout of transsulfuration and TRP14 was non-lethal under basal conditions but lethal under oxidative stress conditions. In the acute pancreatitis model in mice, knockout of TRP14 became protective towards tissue damage in conjunction with a more robust Nrf2 activation than in control animals (lower panels). Created with BioRender.com.

Eksigent, MA, USA) and desalted with 0.1% TFA at 2 µL/min during 10 min. Desalted samples were then loaded into an analytical column (LC Column, 3C18-CL-120, 3 µ, 120 Å 75 µm × 15 cm, Eksigent, MA, USA) equilibrated with 5% acetonitrile 0.1% formic acid. Elution was carried out with a linear gradient of 5 to 35% B in A for 60 min (A: 0.1% formic acid; B: acetonitrile, 0.1% formic acid) at a flow rate of 300 nL/min. Eluted samples were analyzed in a mass spectrometer (5600 TripleTOF, ABSCIEX, MA, USA). Samples were ionized applying 2.8 kV to the spray emitter. Analysis was carried out in a data-dependent mode. Survey MS1 scans were acquired from 350–1250 $m/z$ for 250 ms. The quadrupole resolution was set to 'UNIT' for MS2 experiments, which were acquired at 100–1500 $m/z$ for 50 ms in the high sensitivity mode. The following switch criteria were used: charge: 2+ to 5+; minimum intensity: 70 counts per second. Up to 25 ions were selected for fragmentation after each survey scan. Dynamic exclusion was set to 15 s. The system sensitivity was controlled using 2 fmol of 6 different proteins (LC Packings, CA, USA).

ProteinPilot default parameters were used to generate a peak list directly from 5600 TripleTof wiff files. The Paragon algorithm was used to search the SwisProt database with the following parameters: trypsin specificity and cys-alkylation with taxonomy restricted to human, and false discovery rate (FDR) correction for proteins. Protein grouping was performed using the Pro Group algorithm, which considers protein sets sharing physical evidence. The formation of protein groups in Pro Group was guided only and entirely by observed peptides. Since the observed peptides were actually determined from experimentally acquired spectra, the grouping could be considered to be guided by the spectra. All eluted samples from all experiments were combined to create a spectra library for differential expression analysis. A total of 1441 proteins were identified with 95% confidence and FDR lower than 1% (see dataset in PRIDE PXD050210). Peak View 1.1. software (Sciex, MA, USA) was used to quantify the areas for all the peptides assigned in the library and the computed areas for proteins. Dimensionality reduction, principal component analysis and discriminant analysis (both with Pareto scaling) were carried out for the different experimental groups. A multiple T-test statistical analysis was performed to determine differentially expressed proteins with $P < 0.05$.

## Cysteinylation of recombinant human Prx2 (hPrx2)

Protein cysteinylation was achieved in vitro upon incubation of purified hPrx2 with BODIPY™ FL L-Cystine (ThermoFisher, Massachusetts, USA). hPrx2 samples were pre-reduced with 1 mM DTT for 30 min at room temperature (RT), and then desalted using a Zeba™ Spin Desalting Column, 40 K MWKO, 0.5 mL (ThermoFisher, MA, USA) and incubated with 100 µM BODIPY™ FL L-Cystine overnight at 4 °C protected from light. After incubation, the remaining non-bound BODIPY™ FL L-Cystine was removed by desalting. Protein samples were loaded into a gel (Novex™ 4–20% Tris-Glycine, Invitrogen, CA, USA) using non-reducing loading buffer (NuPAGE™LDS Sample Buffer, Invitrogen, CA, USA) to conserve the mixed disulfide formed in the previous step. Electrophoretic separation was performed at a constant voltage of 200 V for 20–25 min while the gel was protected from light. Once resolved, the gel was developed in a UV transilluminator (ChemiDoc™ XRS+ Imaging System, Bio-Rad,

CA, USA) and Coomassie staining was later performed as loading control.

## Decysteinylation assays

Cell lysates containing cysteinylated proteins were incubated at RT with 20 µM TRP14, 1 µM TrxR1, and 1 mM NADPH in TE buffer (50 mM Tris, 1 mM EDTA). Different combinations of the components of the enzymatic system were tested, as well as different incubation times (0–120 min). After the different incubation times, non-reducing loading buffer (NuPAGE™ LDS Sample Buffer, Invitrogen, CA, USA) was added to the samples and they were heated at 95 °C for 5 min to perform a western blot under non-reducing conditions. Ponceau staining was performed as loading control.

Cysteinylated hPrx2 was incubated at RT with the Trx1 enzymatic system (10 µM Trx1, 30 nM TrxR1, and 300 mM NADPH) or the TRP14 enzymatic system (10 µM TRP14, 30 nM TrxR1, and 300 mM NADPH) in TE buffer (50 mM Tris, 1 mM EDTA). Different Trx1 and TRP14 concentrations, incubation times, and combinations of the components of each reaction system were tested. In addition, the effect of insulin (300 µM) and hydrogen peroxide (100 µM) on the decysteinylation capacity of both systems was also studied. After the incubation, non-reducing loading buffer (NuPAGE™LDS Sample Buffer, Invitrogen, California, USA) was added to samples and they were heated at 95 °C for 5 min before loading them into a gel (Novex™ 4–20% Tris-Glycine, Invitrogen, CA, USA) to perform electrophoresis. Electrophoretic separation was performed at a constant voltage of 200 V for 20–25 min while the gel was protected from light. The gel was developed in a UV transilluminator (ChemiDoc™ XRS+ Imaging System, Bio-Rad, CA, USA), and Coomassie staining was later performed as a loading control.

## Determination of enzyme activities

Prx2 activity was assessed monitoring hydrogen peroxide ($H_2O_2$) clearance from the reaction medium in the presence of the thioredoxin system using the ferrous oxidation-xylenol orange (FOX) assay (Jiang et al, 1992). Two different reaction mixtures were prepared for each sample to assess hPrx2 and cysteinylated hPrx2 activity: 1 µM Prx2 or 1 µM Prx2-S-S-Cys (cysteinylated Prx2), 1 µM Trx1, 2 nM TrxR1, 300 nM NADPH, and 100 mM $H_2O_2$ in TE buffer (50 mM Tris, 1 mM EDTA). At different time points (0–5400 s), 10 µL of the sample were withdrawn from the reaction mixture and added to 190 µL of FOX reagent in a microtiter plate. After 30 min, absorbance was measured at 560 nm in a TECAN Infinity® 200 PRO plate reader (Männedorf, Switzerland). $H_2O_2$ concentration in the reaction medium was calculated by interpolating the absorbance measurements of the sample in a calibration curve containing different concentrations of $H_2O_2$ (0–200 µM). Both calibration curve and sample measurements were run in parallel for each experiment.

α-Amylase and pancreatic lipase activities were measured in serum using the Amylase-LQ and Lipase-HQ detection kits from Spinreact (Barcelona, Spain), according to the manufacturer's instructions. Blood samples were carefully withdrawn with a heparinized syringe from mice under inhalation anesthesia (3%

isoflurane) and centrifuged at 400×g for 15 min at room temperature. Results were expressed as international units per liter of sample (IU/L).

MPO activity determination in pancreatic tissue samples was performed as previously described (Schierwagen et al, 1990). MPO activity was expressed as milliunits per milligram of sample (mU/mg).

## Induction of acute pancreatitis

AP was induced in 12 weeks-old mice by seven intraperitoneal injections of cerulein (Merck, NJ, USA) (50 µg/kg body weight) at 1 h intervals as previously described (Niederau et al, 1985; Pérez et al, 2019). Then, 1 h after the last injection, animals were euthanized under anesthesia with 3% isoflurane, exsanguinated, and the pancreas was immediately removed and processed according to the technique to be used in the different determinations. Blood samples were collected by intracardiac puncture. Death was confirmed by cervical dislocation. 0.9% NaCl solution (B. Braun, Melsungen, Germany) was administered to the control group through seven intraperitoneal injections at hourly intervals. Aminooxacetic acid was dissolved in water, neutralized, diluted into sterile injection-grade saline, and administered intraperitoneally, with the first dose at 15 mg/kg 1 h prior to the first cerulein injection, and then 7 hourly doses were given at 2 mg/kg coincident with the cerulein doses.

Randomization was used to distribute the mice among the different experimental groups. The sample size in mice was calculated considering the mean values and standard deviation of pancreatic GSH levels in wild-type mice under basal conditions or with pancreatitis. Accepting an alpha risk of 0.05 and a beta risk of 0.2 in a two-sided test, five mice are necessary in each group and gender to recognize a statistically significant difference greater than or equal to 3 units. The common standard deviation is assumed to be 1.6. It has been anticipated a drop-out rate of 0%.

## Free and protein-bound low-molecular-weight thiols

Low-molecular-weight thiols and their oxidized disulfide forms as well as mixed disulfides between low-molecular-weight thiols and proteins through disulfide bonds were quantified from pancreatic samples by UHPLC-MS/MS as previously described by us (Moreno et al, 2014).

## Western blot

Tissue samples were homogenized on ice in lysis buffer (100 mg tissue/ 1 mL buffer). The lysis buffer contained 20 mM Tris-HCl (pH 7.5), 1 mM EDTA, 150 mM NaCl, 0.1% SDS, 1% Igepal, 30 mM sodium pyrophosphate, 50 mM sodium fluoride and 50 µM sodium vanadate. A protease inhibitor cocktail (Sigma-Aldrich, MO, USA) was added at a concentration of 10 µL/mL before homogenization.

Cell lysates were obtained by adding 80 µL of cOmplete™ Lysis-M buffer supplemented with an EDTA-free protease inhibitor cocktail (Roche, Basel, Switzerland) per well and scraping the cells with a disposable cell scraper.

Then, 10.5 µL of protein samples were mixed with 3.5 µL of NuPAGE™ LDS Sample Buffer (ThermoFisher, MA, USA) and loaded into a gradient gel ranging from 4–20% acrylamide (Bio-Rad, CA, USA). When electrophoresis was performed under reducing conditions, DTT was added to the loading buffer to give a final concentration of 100 nM.

A constant-voltage electric field was applied to the gel (120 volts) through a Tris-glycine running buffer (25 mM Tris, 200 mM glycine, 0.1% SDS, pH 8.3). The resolved gel was transferred to a Trans-Blot™ Turbo™ Midi Nitrocellulose transfer pack (Bio-Rad, CA, USA). Semi-dry electrotransference was performed using the Trans-Blot™ Turbo™ transfer system (Bio-Rad, CA, USA). Membranes were then incubated for 1 h in the blocking buffer to avoid nonspecific antibody binding. Blocking buffer consisted in 5% bovine serum albumin in T-TBS (0.1% Tween-20, 20 nM Tris, 137 mM NaCl, pH 7.6). Regarding cystathionine β-synthase (CBS) blots, the membrane was incubated with the primary antibody (AB140600, Abcam, Cambridge, UK) overnight, and the membrane was then incubated with the secondary antibody (anti-rabbit IgG-HRP, Cell Signaling, MA, USA) for 1 h at RT. For detecting biotin-labeled cysteinylated proteins, membranes were incubated for 1 at RT with streptavidin-HRP (Cell Signaling, MA, USA). Membranes were then incubated with SuperSignal™ West Dura Extended Duration Substrate (Thermo Scientific, Waltham, USA) for 5 min before being developed in the ChemiDoc™ XRS+ Imaging System™ (Bio-Rad, CA, USA).

## Quantitative real-time PCR (RT-qPCR)

To extract and isolate RNA from the pancreas, the tissue was stored in RNAlater™ solution (Invitrogen, CA, USA) and processed within 24 h to prevent RNA degradation by RNAses present in the pancreas. 25–30 mg of pancreatic tissue were homogenized in 500 µL of TRI Reagent™ (Sigma-Aldrich, MO, USA). Isolation was performed following the manufacturer's instructions.

Once RNA was isolated, reverse transcription was performed to obtain the complementary DNA sequences using PrimeScript™ RT Reagent Kit (Takara, Shiga, Japan) according to the manufacturer's instructions in a C1000 Thermal Cycler (Bio-Rad, CA, USA). RT-qPCR was carried out using an iQ™5 Multicolor Real-Time PCR Detection System (Bio-Rad, CA, USA) thermal cycler coupled to fluorescence detection. Every reaction was performed in triplicate and melt curves were analyzed with iQ™ Real Time Detection System Software (Bio-Rad, CA, USA) to check that only one PCR product per sample was formed. Oligonucleotide sequences for each specific gene were synthetized by Sigma-Aldrich (MO, USA) (Appendix Table S6). The master mix used for PCR was TB Green™ Premix Ex Taq™ (Tli RNase H Plus) (Takara, Shiga, Japan) which contains polymerase, nucleotides, and fluorescent dye, which were needed to perform the reaction. *TATA-box-binding protein (Tbp)* was used as the housekeeping gene. The thermal cycler was set according to the following amplification scheme: 10 min at 95 °C, 40 cycles of 15 s at 95 °C followed by 30 s at 60–64 °C for annealing (according to the optimum hybridization temperature for each primer set) and 30 s at 72 °C for elongation.

Relative mRNA expression was calculated taking into account the threshold cycle from each gene according to the following formula: relative expression = $2^{-\Delta(\Delta CT)}$.

## Proteomic analysis of pancreatic samples

### In-gel digestion of proteins

The protein extracts were suspended in a volume up to 50 µl of sample buffer, and then applied onto 1.2-cm wide wells of a conventional SDS-PAGE gel (0.75 mm-thick, 4% stacking, and 10% resolving). The run was stopped as soon as the front entered 3 mm

into the resolving gel, so that the whole proteome became concentrated in the stacking/resolving gel interface. The unseparated protein bands were visualized by Coomassie staining, excised, cut into cubes (2 × 2 mm), and placed in 0.5 ml microcentrifuge tubes (Moreno et al, 2014). The gel pieces were destained in acetonitrile:water (ACN:H$_2$O, 1:1) and digested in situ with sequencing-grade trypsin (Promega, Madison, WI) as described by Shevchenko et al (Shevchenko et al, 2006), with minor modifications. The gel pieces were shrunk by removing all liquid using sufficient ACN. Acetonitrile was pipetted out and the gel pieces were dried in a speedvac. The dried gel pieces were re-swollen in 100 mM Tris-HCl pH 8, 10 mM CaCl$_2$ with 60 ng/μl trypsin at 5:1 protein:enzyme (w/w) ratio. The tubes were kept in ice for 2 h and incubated at 37 °C for 12 h. Digestion was stopped by the addition of 1% TFA. Whole supernatants were dried down and then desalted onto OMIX Pipette tips C18 (Agilent Technologies) until the mass spectrometric analysis.

### Reverse phase-liquid chromatography and RP-LC-MS/MS analysis

The desalted protein digest was dried, resuspended in 10 μl of 0.1% formic acid, and analyzed by RP-LC-MS/MS in an Easy-nLC II system coupled to an ion trap LTQ-Orbitrap-Velos-Pro hybrid mass spectrometer (Thermo Scientific). The peptides were concentrated (online) by reverse phase chromatography using a 0.1 mm × 20 mm C18 RP precolumn (Thermo Scientific), and then separated using a 0.075 mm × 250 mm C18 RP column (Phenomenex) operating at 0.22 μl/min. Peptides were eluted using a 180-min dual gradient. The gradient profile was set as follows: 5–25% solvent B for 135 min, 25–40% solvent B for 45 min, 40–100% solvent B for 2 min and 100% solvent B for 18 min (Solvent A: 0,1% formic acid in water, solvent B: 0,1% formic acid, 80% acetonitrile in water). ESI ionization was done using a Nano-bore emitters Stainless Steel ID 30 μm (Proxeon) interface at 2.1 kV spray voltage with S-Lens of 60%. The Orbitrap resolution was set at 30.000 (Alonso et al, 2015).

Peptides were detected in survey scans from 400 to 1600 amu (1 μscan), followed by 20 data-dependent MS/MS scans (Top 20), using an isolation width of 2 u (in mass-to-charge ratio units), normalized collision energy of 35%, and dynamic exclusion applied during 60 s periods. Charge-state screening was enabled to reject unassigned and singly charged protonated ions.

### Data processing and quantitation of peptides

Peptide identification from raw data (Technical triplicates) was carried out twice using PEAKS Studio XPro search engine (Bioinformatics Solutions Inc., Waterloo, ON, Canada) (Tran et al, 2016, 2017, 2019). Database search was performed against uniprot-mus-musculus.fasta (55474 entries; UniProt release 10/2020) (decoy-fusion database). The following constraints were used for the searches: tryptic cleavage after Arg and Lys (semispecific), up to two missed cleavage sites, and tolerances of 20 ppm for precursor ions and 0.6 Da for MS/MS fragment ions and the searches were performed allowing optional Met oxidation, N-ethylmaleimide on Cys and Cysteinylation. False discovery rates (FDR) for peptide spectrum matches (PSM) and for protein were limited to 0.01. Only those proteins with at least two unique peptides being discovered from LC/MS/MS analyses were considered reliably identified and sent to be quantified.

Quantitation of peptides was performed with PEAKS Studio XPro search engine, selected "Label Free Quantification" under the "Quantifications" options using 20 ppm for mass error tolerance and 3 min for retention time shift tolerance (Clement et al, 2018). We use the total ion current (TIC) of the samples to calculate the normalization factors. Normalized abundance is calculated from the raw abundance divided by the normalization factor. The Quality (10–13) and Avg. Intensity (6e4) were used for Spectrum filter and Significance (20, ANOVA method) was used for peptide and protein abundance calculation. For protein quantification, we considered protein groups for peptide uniqueness, only unique peptides were used for protein quantification and the modified peptides were excluded. The mass spectrometry proteomics data have been deposited in the PRIDE partner repository with the dataset identifier PXD050765.

### Histological analysis

Pancreas were removed and fixed in freshly prepared 4% paraformaldehyde (PFA). After 24 h of fixation in PFA, the samples were sent to the Microscopy Section from the SCSIE from the Universitat de València for paraffin inclusion. Then, 5-μm-thick tissue sections were cut and stained with hematoxylin–eosin. At least 20 fields per tissue sample were observed under the brightfield microscope. The extent and severity of tissue edema and inflammatory infiltrate were assessed blindly using a scale ranging from 0 to 3 as previously described (Van Laethem et al, 1998), observing pancreas sections stained with hematoxylin–eosin.

### Immunohistochemistry staining

After deparaffinization and rehydration, pancreas sections were subjected to enzymatic antigen retrieval with citrate buffer (0.1 M, pH 6.0) for 40 min at 94 °C. After cooling, they were permeabilized with GS-PBS-T solution (goat serum 1%, Triton 0.4%) and blocked with GS 3.5%-PBS-T for 1 h. After blocking, the samples were incubated overnight with the primary antibody (anti-CD11b/ Integrin Alpha M, at 25 μg/mL, R&D Systems) prepared in GS 3.5%-PBS-T. Samples were then incubated with secondary antibody (Alexa 488, ThermoFisher) and Hoechst (1:1000, Invitrogen) for 1 h. FluorSave™ reagent (Millipore) was used as mounting medium.

### Statistical analysis

The results were expressed as a mean ± standard deviation. First, two-way ANOVA was performed to compare the mean of the different groups. Then, when significant differences were observed, the difference between the individual groups was determined using either Tukey's test or t-test; the statistical analysis performed in each case is mentioned in the Figure legends. The analyses in which a $P < 0.05$ was obtained were considered significant.

# Data availability

The mass spectrometry proteomics data have been deposited to the ProteomeXchange Consortium via the PRIDE partner repository (https://www.ebi.ac.uk/pride/) with the dataset identifiers PXD050210, PXD050368, and PXD050765.

The source data of this paper are collected in the following database record: biostudies:S-SCDT-10_1038-S44318-024-00117-1.

## Peer review information

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

## Acknowledgements

The TRP14 knockout HEK293 cells were made by the Karolinska Genome Engineering (KGE) core facility. Some *C. elegans* strains were provided by the Caenorhabditis Genetics Center (CGC), which is funded by NIH Office of Research Infrastructure Programs (P40 OD010440) USA. We thank SunyBiotech (https://www.sunybiotech.com/) for their excellent assistance in generating the PHX4767, *txdc-17(syb4767)* strain by CRISPR-Cas9 and Profs. Milos R Filipovic and Simon Tuck for providing strains. We thank Luz Valero from the Proteomics Section of the Central Service for Experimental Research (SCSIE) from the Universitat de València for the proteomic analysis of HEK293 cells and peroxiredoxin 2. JS acknowledges funding by grants from the Spanish Government (PID2019-108615RB-I00 from the Agencia Estatal de Investigación (AEI); and SAF2015-71208-R from the Ministry of Economy and Competitiveness) both with funds from the European Regional Development Fund (ERDF) from the European Union (EU). JS and AMR also acknowledge scientific advice and guidance provided by the Spanish Research Biology and Medicine Redox Network RED2018-102576-T. ESJA acknowledges funding from Karolinska Institutet, The Knut and Alice Wallenberg Foundations (KAW 2019.0059), The Swedish Cancer Society (21 1463 Pj), The Swedish Research Council (2021-02214), National Laboratories Excellence program under the National Tumor Biology Laboratory project (2022-2.1.1-NL-2022-00010) and the Hungarian Thematic Excellence Programme (TKP2021-EGA-44) and The National Research, Development and Innovation Office (NKFIH) grant ED_18-1-2019-0025. EES acknowledges funding from the United States National Institutes of Health grants AG040020, AG055022, OD026444, DK123738, and P30GM140964, as well as from the Hungarian Eötvös Loránd Kutatási Hálózat Foundation (ELKH) and the Hungarian Magyar Tudományos Akadémia (MTA, grant K-22 #143769). PN acknowledges funding from the National Research, Development and Innovation Fund of the Ministry of Culture and Innovation under the National Laboratories Program, National Tumor Biology Laboratory grant 2022-2.1.1-NL-2022-00010, the National Research, Development and Innovation Office under the Hungarian Thematic Excellence Program grant TKP2021-EGA-44. Furthermore, PN and EES acknowledge the HUN-REN Hungarian Research Network grant 1500207 and a Distinguished Guest Fellowship from the Hungarian Academy of Sciences (#AT02023-26). KB-G acknowledges funding from the Richter Gedeon Talentum Foundation, founded by Richter Gedeon Plc. HCL, DGG, and AMV were supported by Projects PRE2019-088198, PGC2018-094276-B-I00, and PID2021-122311NB-I00 from the Spanish Government AEI, and P20_00229 from the Conserjería de Economía, Conocimiento, Empresas y Universidad, Junta de Andalucía), all with funds from the EU ERDF. RL was supported by Project PID2021-123481OB-I00 from the Spanish Government AEI. The proteomic analysis in pancreatic samples was carried out in the 'CBMSO PROTEIN CHEMISTRY FACILITY', which belongs to ProteoRed, PRB3-ISCIII, supported by grant PT17/0019/0018 from the Instituto de Salud Carlos III (ISCIII) of the Spanish Government, with funds of the ERDF. AMR acknowledges funding from the Spanish Government grants RTI2018-094203-B-I00 and PID2021-124688OB-I00 from the AEI, with funds of the EU ERDF.

## Author contributions

**Pablo Martí-Andrés**: Data curation; Software; Formal analysis; Validation; Investigation; Methodology; Writing—original draft. **Isabela Finamor**: Data curation; Software; Formal analysis; Investigation; Methodology. **Isabel Torres-Cuevas**: Supervision; Investigation; Methodology. **Salvador Pérez**: Data curation; Formal analysis; Methodology. **Sergio Rius-Pérez**: Data curation; Formal analysis; Methodology. **Hildegard Colino-Lage**: Data curation; Formal analysis; Investigation; Methodology. **David Guerrero-Gómez**: Data curation; Formal analysis; Investigation; Methodology. **Esperanza Morato**: Data curation; Formal analysis; Validation; Investigation; Methodology. **Anabel Marina**: Data curation; Formal analysis; Validation; Investigation; Methodology. **Patrycja Michalska**: Data curation; Formal analysis; Methodology. **Rafael León**: Data curation; Formal analysis; Supervision; Investigation; Methodology. **Qing Cheng**: Data curation; Formal analysis; Methodology. **Eszter Petra Jurányi**: Data curation; Formal analysis; Methodology. **Klaudia Borbényi-Galambos**: Data curation; Formal analysis; Methodology. **Iván Millán**: Data curation; Formal analysis; Methodology. **Péter Nagy**: Supervision; Investigation; Methodology; Writing—review and editing. **Antonio Miranda-Vizuete**: Conceptualization; Data curation; Formal analysis; Supervision; Funding acquisition; Investigation; Methodology; Writing—review and editing. **Edward E Schmidt**: Conceptualization; Supervision; Funding acquisition; Investigation; Methodology; Writing—review and editing. **Antonio Martínez-Ruiz**: Supervision; Validation; Investigation; Methodology. **Elias SJ Arnér**: Conceptualization; Resources; Formal analysis; Supervision; Funding acquisition; Validation; Investigation; Methodology; Writing—original draft; Writing—review and editing. **Juan Sastre**: Conceptualization; Formal analysis; Supervision; Funding acquisition; Validation; Investigation; Methodology; Writing—original draft; Writing—review and editing.

Source data underlying figure panels in this paper may have individual authorship assigned. Where available, figure panel/source data authorship is listed in the following database record: biostudies:S-SCDT-10_1038-S44318-024-00117-1.

## Funding

## Disclosure and competing interests statement

The authors declare no competing interests.

# Expanded View Figure

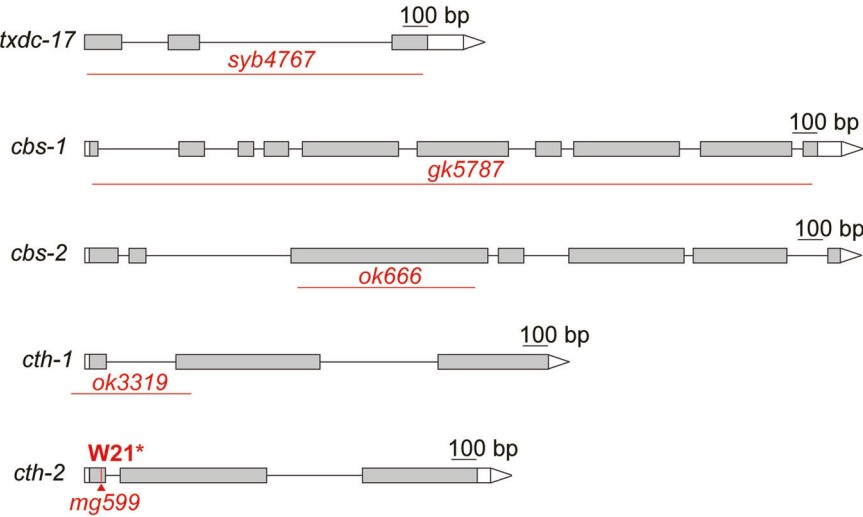

**Figure EV1.  Schematic representation of the genes and alleles in *C. elegans* used in this study.**

Grey boxes indicate exons encoding the ORF and white boxes represent the UTRs. The molecular lesions are depicted in red.

