## [Peer Review File · The EMBO Journal]

TRP14 is the rate-limiting enzyme for intracellular cystine reduction and regulates proteome cysteinylation

Pablo Martí-Andrés, Isabela Finamor, Isabel Torres, Salvador Pérez, Sergio Rius-Pérez, Hildegard Colino-Lage, David Guerrero Gomez, Esperanza Morato, Anabel Marina, Patrycja Michalska, Rafael León, Qing Cheng, Eszter Jurányi, Klaudia Borbényi-Galambos, Ivan Millán, Péter Nagy, Antonio Miranda Vizueté, Ed Schmidt, Antonio Martínez-Ruiz, Elias Arnér, and Juan Sastre

Corresponding author(s): Juan Sastre (juan.sastre@uv.es) , Elias Arnér (Elias.Arner@ki.se)

Review Timeline:

Submission Date:	17th Apr 23
Editorial Decision:	25th May 23
Revision Received:	25th Feb 24
Editorial Decision:	15th Apr 24
Revision Received:	26th Apr 24
Accepted:	26th Apr 24

Editor: Hartmut Vodermaier

Transaction Report:

Prof. Juan Sastre
University of Valencia, Faculty of Pharmacy
Department of Physiology
Avda. Vicente Andrés Estellés s/n, School of Pharmacy
Burjasot, Valencia 46100
Spain

25th May 2023

Re: EMBOJ-2023-114299
TRP14 is the rate limiting enzyme for intracellular cystine reduction and regulates proteome cysteinylolation

Dear Dr. Sastre,

Thank you for submitting your manuscript on TRP14 and cellular cystine reduction to The EMBO Journal. I have now heard back from three expert referees, whose reports are copied below. As you will see, all referees consider your study and its findings potentially interesting. However, while referees 1 and 3 raise only a number of specific concerns to be addressed, reviewer 2 remains on the whole unconvinced by the main conclusions of the study, and would require decisive additional evidence in their support.

Should you be able to satisfactorily address these criticisms with additional complementary experiments and controls, we would be interested in pursuing a revised version of the study further for publication. However, since it is our policy to consider only a single round of major revision and therefore important to fully answer to all comments at the time of resubmission, I would invite you to get back to me with a tentative response letter/revision plan already during the early stages of the revision work. On the basis of this response, we could then further discuss the revision requirements and how to best address the key concerns e.g. via email or a follow-up video call. I should add that we could also offer extension of the default three-months revision period if needed. Our 'scooping protection' (meaning that competing work appearing elsewhere in the meantime will not affect our considerations of your study) will remain valid throughout the revision, even if extended.

Detailed information on preparing, formatting and uploading a revised manuscript can be found below and in our Guide to Authors. Thank you again for the opportunity to consider this work for The EMBO Journal, and I look forward to hearing from you in due time.

Yours sincerely,

Hartmut Vodermaier

3) Revised manuscript text (including main tables, and figure legends for main and EV figures) has to be submitted as editable

text file (e.g., .docx format). We encourage highlighting of changes (e.g., via text color) for the referees' reference.

4) Each main and each Expanded View (EV) figure should be uploaded as individual production-quality files (preferably in .eps, .tif, .jpg formats). For suggestions on figure preparation/layout, please refer to our Figure Preparation Guidelines:

8) Please note that supplementary information at EMBO Press has been superseded by the 'Expanded View' for inclusion of additional figures, tables, movies or datasets; with up to five EV Figures being typeset and directly accessible in the HTML version of the article. For details and guidance, please refer to:

embopress.org/page/journal/14602075/authorguide#expandedview

9) Digital image enhancement is acceptable practice, as long as it accurately represents the original data and conforms to community standards. If a figure has been subjected to significant electronic manipulation, this must be clearly noted in the figure legend and/or the 'Materials and Methods' section. The editors reserve the right to request original versions of figures and the original images that were used to assemble the figure. Finally, we generally encourage uploading of numerical as well as gel/blot image source data; for details see: embopress.org/page/journal/14602075/authorguide#sourcedata

At EMBO Press, we ask authors to provide source data for the main manuscript figures. Our source data coordinator will contact you to discuss which figure panels we would need source data for and will also provide you with helpful tips on how to upload and organize the files.

Further information is available in our Guide For Authors:

In the interest of ensuring the conceptual advance provided by the work, we recommend submitting a revision within 3 months (23rd Aug 2023). Please discuss the revision progress ahead of this time with the editor if you require more time to complete the revisions. Use the link below to submit your revision:

Link Not Available

Referee #1:

Cysteine is an important amino acid, not only for its role as constituent of proteins but also as building block of glutathione or sulfur source for the synthesis of iron-sulfur clusters. Cysteine is taken up by a dedicated antiporter system in the plasma membrane in its oxidized form as cystine. For further use, it thus has to be reduced. While over the years many reducing systems have been implicated in this process, the major cystine reductase has remained unknown.

In this manuscript, the authors propose TRP14 as the major and rate-limiting cystine reductase in cells. They demonstrate that the transsulfuration pathway becomes in TRP14 knockout models the major source of cysteine and they put their findings into context in a *C. elegans* and mouse disease model. The authors provide a wide array of data, many of which are of high quality that support their conclusions.

My major concern is the following: I like the principal punch line of this manuscript (TRP14 is limiting in cells for cystine reduction and the resulting physiological consequences). However, the manuscript "feels" stitched together. Data from cell culture (KO and knockdown experiments), in vitro reconstituted cell lysate experiments, *C. elegans* and a mouse disease model are presented without a clear and logical connection. In each of the systems, the respective data sets appear somewhat preliminary and incomplete. Since in their previous PNAS manuscript on the matter (Pader PNAS 2014), the authors already extensively characterized the cystine reduction capacity of TRP14 in vitro and in lysates of TRP14-depleted cells, a stronger focus on the organismal models would make this manuscript more convincing and better readable.

Other points are:

1/ Fig. 1A: I would expect to see complementation experiments of the TRP14 KO cells with WT and active site mutant of TRP14.

2/ Why did the authors not measure total cysteine/cystine/glutathione etc levels in the HEK293 TRP14 knockout cells?

3/ Nrf2 is strongly induced upon treatment of TRP14 KO cells with Cystine. Are Nrf2 targets also induced?

4/ C. elegans and mouse data indicate a stronger use of the transsulfuration pathway. Is this also the case in their cellular models? Did the flux through the transsulfuration pathway change in the different models? Did the amounts of involved enzymes change?

5/ Fig. 2: the author analysed here TRP14 knockdown cells. Why did they not use their knockout cells?

6/ In TRP14 KD cells protein cysteinylolation is increased upon treatment with CysBio. In the mouse model, induction of oxidative stress results in lower amounts of protein cysteinylolation. An insightful experiment might be to test in cells how quickly cysteinylolation is removed from proteins and whether in this respect in TRP14 knockout cells faster removal kinetics can be observed. This could help to bring both datasets together conceptually. How efficient are the Trx1 or the glutaredoxin systems in reverting protein-cysteinylolation?

7/ Fig. 2B: A complementation with the active-site mutant of TRP14 and the WT would be required.

8/ The TRP14 knockout mice were protected from tissue damage in acute pancreatitis. I lack here more mechanistic insights explaining the "why". The authors propose an "over-compensation" by "Nrf2 activation and upregulation of transsulfuration", but only provide descriptive data.

9/ I could not find any data availability statement. All proteomic data must be uploaded to respective databases

Minor

1/ size bar indicating the dimensions of the microscopic images is missing

2/ labelling/font size etc of the images is different between panels (compare e.g. 1B with 1C); compare Fig. 3 and 4 -AP/CT vs control/pancreatitis?

3/ I am surprised that the authors did not test the susceptibility of the TRP14 KO cells to ferroptosis (inhibition of system XC-faithfully leads to ferroptosis induction)

4/ Are their changes in TRP14 levels/activity under disease conditions?

Referee #2:

This is an interesting manuscript proposing that TRP-14 is the major mechanism by which cystine disulfide (cystine) is reduced to the thiol state in cells. The authors also provide data about the changes of cystine metabolism in WT or TRP-14 null mice during pancreatitis - but I found this confusing as I will articulate below.

Many of the Figures are difficult to read with the text being small - well below publication quality and hinders the review process and the authors communication of their data. There are no Figure numbers.

The authors use BODIPY FL L-Cystine. In Fig 1A the work use BODIPY FL is used, but this is unhelpful because the crucial fact is that this relates to cystine (arguably the most important point) and this is not immediately clear. More importantly, I was surprised the authors were able to use this as a selective substrate for TRP14, which is what might be taken from Fig 1A as TRP14 KO reduced the signal to essentially zero. This reagent (<https://www.thermofisher.com/order/catalog/product/B20340>) has been used to index total thiol because as it says in the manufacturers information, it thiol-disulfide exchanges to liberate cysteine-FL that fluoresces. A living cell will be reducing inside, and this will include TRP-15 null cells as having predominantly oxidised thiols is not compatible with life. Therefore, it is extremely difficult to accept that the null cells cannot reduce this dye. Of course, the authors may have set a threshold that gives the numbers they report, but the problem is this is not absolutely quantitative and is instead arbitrary units and relative to WT. This can be done truly quantitatively (i.e., with proper units on the y axis) in cell lysates, which is important to address the point that it currently appears that the BODIPY FL L-Cystine is not reducible in the absence of TRP-14. All manner of thiol will reduce it.

A major limitation is that the authors do not quantify the contribution of TRP-14 to cystine reduction. They quickly move on to Fig 1B with various manipulations before pinning down extremely well that TRP-14 is the principal reducer of cystine. Even in the experiment in Fig 1C showing TRP-14 can reduce cystine, this is not really unexpected. What about other thiol reducing

enzymes - they probably do as well in this assay system. Comparisons with other thiol reducing system, with rates of reduction and enzyme kinetics is warranted given the claims made.

Fig 2A/B - there are no loading controls shown. In Fig 2B, I am sure that any / many thiol reducing system added with decrease the signal and a range of other reducing enzymes would ideally have been compared and there may have also been a control with the TRP-14 - as feeding NADPH or NADPH & TRxR may be expected to reduce the signal. Control for incubation times is missing too. The Prx reactivation experiments in 2E - again, probably any thiol reductant system added to the inactivated Prx will achieve the same. Perhaps TRP-14 is much 'better' at doing this, but there needs to be some comparisons and kinetics.

The studies comparing WT with TRP-14 (Txndc17) knock-out mice are interesting, but the choice of performing these experiments is not very clear and the connectivity with the preceding work is questionable, especially as the first part on cystine reduction by TRP-14 is not very compelling. I guess it an attempt to bring a biomedical relevance of the attributed TRP-14 selective reduction of cystine - but is not only premature in my opinion because the TRP-14 selective reduction of cystine is not robust, but because there is no clear mechanistic explanation for the observations made in the pancreatitis model. The biological observations in pancreatitis model have intrinsic value, but mechanistically it does not connect well with the initial focus. Of course, the measurements of thiol redox state in the tissue relate to the first part of the study, but this is a complex model. It doesn't make sense to me to try and use this as a demonstration that a failure of TRP-14 to reduce cystine has a biological consequence using a model with such complexity. Why does the null mouse not have elevated cystine basally?

Overall, I am not convinced that that TRP-14 is a highly selective cystine reductase and that this protein has a crucial role to play in the pathogenesis of pancreatitis. There are some notable differences in the responses of the null mouse to the disease model compared to WT, but whether this can be attributed to a failure to reduce cystine is not proven in my opinion and therefore the observations are really just detailed phenotyping without bringing robust mechanistic understanding.

Referee #3:

This manuscript addresses the question of what happens to cystine taken up by cells and how it is reduced to cysteine such that it can be fed into biosynthetic processes. This seems to be quite a fundamental question in cell biology, and the authors provide convincing evidence that TRP14 (TXND17) is responsible. The experiments are comprehensive, and the results are convincing.

The comments this reviewer has are minor and mostly textual:

Line 118-119: "We next attempted to block the nematode cystine reduction pathway by generating a txdc-17(syb4767) null allele by CRISPR-Cas9 technology." It would be better if this sentence could avoid the assumption that txdc-17 in worms would be on the cystine reduction pathway (and let it emerge as an outcome of the experiments). For example, something like, "To investigate the role of *C. elegans* TRP14 in vivo, we generated..."

Line 121: "viable and indistinguishable from wild type"

The authors should state in the main text by what parameters the worms were indistinguishable (growth and viability). Presumably there is some other analysis that could be done that would distinguish the txdc-17 worms from wild type.

Lines 121-127: The description of the mutants is a bit confusing. It is stated that first the double mutant *cbs-1* and *cbs-2* was made, but the next sentence just reports the growth of *cbs-1*. Also, it is not clear what is the connection between "Together with the fact that *cbs-2* has been proposed to be a pseudogene" and the decision to impair the transsulfuration pathway at a different level. Even if *cbs-2* were not a pseudogene, it seems that the authors would have had to use another strategy to affect the transsulfuration pathway.

Line 152: There should be a comma after "protein cysteinylolation" in the subheading.

Line 174: This reviewer does not understand Fig. 2C. How does this heat map show higher cysteinylolation levels in TRP14 knockdown cells for all the indicated proteins?

Lines 301-305: This sentence is a bit convoluted.

Line 370: "HEK203" is presumably a typo.

Lines 423-434: Reference 14 is given for a description of Txndc17 mice, but that reference seems only to describe knockdown in cells. Is reference 15 the correct one?

There are spelling and formatting mistakes in the methods section. Here are a few examples:

Line 442: "recrystalized"

Line 552: "H2O2"

Line 595: "scrapping the cells with a disposable cell scrapper" (it should presumably be "scraping the cells with a disposable cell scraper.")

POINT-BY-POINT REPLIES to REVIEWERS of MANUSCRIPT EMBOJ-2023-114299

Referee #1:

Cysteine is an important amino acid, not only for its role as constituent of proteins but also as building block of glutathione or sulfur source for the synthesis of iron-sulfur clusters. Cysteine is taken up by a dedicated antiporter system in the plasma membrane in its oxidized form as cystine. For further use, it thus has to be reduced. While over the years many reducing systems have been implicated in this process, the major cystine reductase has remained unknown.

In this manuscript, the authors propose TRP14 as the major and rate-limiting cystine reductase in cells. They demonstrate that the transsulfuration pathway becomes in TRP14 knockout models the major source of cysteine and they put their findings into context in a *C. elegans* and mouse disease model. The authors provide a wide array of data, many of which are of high quality that support their conclusions.

My **major concern** is the following: I like the principal punch line of this manuscript (TRP14 is limiting in cells for cystine reduction and the resulting physiological consequences). However, **the manuscript "feels" stitched together. Data from cell culture (KO and knockdown experiments), in vitro reconstituted cell lysate experiments, *C. elegans* and a mouse disease model are presented without a clear and logical connection. In each of the systems, the respective data sets appear somewhat preliminary and incomplete.** Since in their previous PNAS manuscript on the matter (Pader PNAS 2014), the authors already extensively characterized the cystine reduction capacity of TRP14 in vitro and in lysates of TRP14-depleted cells, **a stronger focus on the organismal models would make this manuscript more convincing and better readable.**

Answer: We highly appreciate the positive comments of the reviewer on our findings that support the key role of TRP14 as the major and rate-limiting cystine reductase in cells, as well as the important role of the transsulfuration pathway as the major source of cysteine in TRP14 knockout models. With regards to our earlier findings as reported in PNAS 2014, those studies were assessing TRP14-catalyzed cystine reduction *in vitro*, but not in live cells nor in animals. In fact, we were very surprised by our finding that although the TRP14 knockout mice appear without any overt phenotype when not being stressed they became *more resistant* to pancreatitis compared to wildtype controls. This finding spurred us to control for the functions of TRP14 in both human HEK293 cells, *C. elegans*, and in mice, which is the historical background for the layout of our present study. We find that the conclusions we present are both important and solid, duly backed up by our findings in these different model systems. Regarding the comment about an apparent lack of a clear and logical connection between the data from cell culture, in vitro reconstituted cell lysate experiments, *C. elegans*, and the mouse disease model we have now put an emphasis on explaining and introducing the models better in the revised paper and we hope that the Reviewer will find our revised text to address this point in a better and more readable manner.

Other points are:

1/ Fig. 1Aff: I would expect to see **complementation experiments of the TRP14 KO cells with WT and active site mutant of TRP14.**

Answer: Thank you for this excellent suggestion. In response we constructed plasmids to be able to include such additional complementary experiments, which have now been performed. These additional results are included in the new Figures 1D-E and clearly further support the overall findings in this study.

2/ Why did the authors not **measure total cysteine/cystine/glutathione** etc levels in the **HEK293 TRP14 knockout cells?**

Answer: Thank you very much for the suggestion. At normal culturing conditions (200 μ M cystine in the culture medium), we found no significant differences in GSH and cystine levels between parental HEK293 and TRP14 knock-out HEK293 cells, but a decrease in cysteine levels in TRP14 knock-out HEK293 cells as compared to parental HEK293 cells (Appendix Fig. S1A).

3/ Nrf2 is strongly induced upon **treatment of TRP14 KO cells with Cystine. Are Nrf2 targets also induced?**

Answer: The up-regulation of Nrf2 downstream targets has now been shown *in vivo* in TRP14 KO mice in response to acute pancreatitis (Fig. 8H-J), and previously we demonstrated that the activation of Nrf2 detected by pTRAF was accompanied by upregulation of Nrf2 downstream target genes (Kipp et al. Redox Biology 2017). Fig. 3 uses the pTRAF system to show activation of an Nrf2 response in response to treatment with 250 mM cystine to induce oxidative stress, which was stronger in TRP14-KO HEK cells than in WT HEK control cells. Importantly, in both cases, Nrf2 is not induced in the basal TRP14-null state, but it is more strongly induced by either high cystine in the cell system or by acute pancreatitis in the mouse model. We hope that the Reviewer will agree with us that this is a robust Nrf2 response that has been measured both based on Nrf2 activity and based on the expression of multiple classic Nrf2-response genes.

4/ **C. elegans and mouse data indicate a stronger use of the transsulfuration pathway. Is this also the case in their cellular models? Did the flux through the transsulfuration pathway change in the different models? Did the amounts of involved enzymes change?**

Answer: Very good point, thank you. To address this comment the flux through the transsulfuration pathway has now been assessed in both WT and TRP14 KO cells (see on page 6, paragraphs 2 and 3, lines 130-147, and Fig. 2). This was a major undertaking requiring significant additional experimentation and expertise. We thus contacted Dr. Peter Nagy, who has the capability of performing such experiments, and he and his group completed these additional experiments. He is thus now together with two of his group members (Eszter Petra Jurányi and Klaudia Borbényi-Galambos) also included as additional coauthors on the revised manuscript.

5/ **Fig. 2:** the author analysed here TRP14 knockdown cells. **Why did they not use their knockout cells?**

Answer: These experiments on protein cysteinylolation were performed at the beginning of our study, when only TRP14 knockdown cells were available. Later, we developed TRP14 knockout cells and confirmed that the profile of TRP14 and thioredoxin 1 protein levels were very similar between TRP14 knockout and knockdown cells, which we therefore used for the remaining studies. Having both sets of experiments included in the paper further solidifies it, we believe.

6/ In TRP14 KD cells protein cysteinylolation is increased upon treatment with CysBio. In the mouse model, induction of oxidative stress results in lower amounts of protein cysteinylolation. An insightful experiment might be **to test in cells how quickly cysteinylolation is removed from proteins and whether in this respect in TRP14 knockout cells faster removal kinetics can be observed**. This could help to bring both datasets together conceptionally.

How efficient are the Trx1 or the glutaredoxin systems in reverting protein-cysteinylolation?

Answer: The efficiency of the TRP14, Trx1, glutaredoxin, and glutathione systems in reverting protein-cysteinylolation was now tested *in vitro* and is shown in the revised version of the manuscript in Figure 5 and in Appendix Figure S3 (some of these experiments were previously also found in the supplementary material of the original paper). It should however be noted that the efficiency *in vitro* is not necessarily reflective of the activities in a cellular context, which we have now noted in the revision. As based upon our results with the live HEK293 cells, *C. elegans* and the mice, we are confident that TRP14 is the rate limiting enzyme for cystine reduction in the cellular context, although most Trx-fold proteins have the capacity to reduce cystine *in vitro*. This statement is now better and more explicitly spelled out in the revised manuscript.

7/ **Fig. 2B: A complementation with the active-site mutant of TRP14 and the WT would be required.**

Answer: Thank you for this good comment. As discussed under the first comment, we complemented the cellular activity measurements using transfections of the wildtype and active site mutant of TRP14 (Fig. 1 D-E), which we believe was an excellent and important control experiment. For *in vitro* experiments, this should be of less novelty and we have not had the time to express new recombinant protein with the active site mutant for these experiments.

8/ **The TRP14 knockout mice were protected from tissue damage in acute pancreatitis. I lack here more mechanistic insights explaining the "why".** The authors propose an **"over-compensation" by "Nrf2 activation and upregulation of transsulfuration"**, but **only provide descriptive data**.

Answer: This is an insightful question about a very complicated physiological response which, of course, will require additional future studies to fully characterize. However, to provide more mechanistic insights into the protection, as requested by the Reviewer, we have performed some key additional experiments on the mouse model using aminooxyacetic acid (AOAA), which inhibits both cystathionine β -synthase (CBS) and cystathionine γ -lyase (CSE), two key enzymes of the transsulfuration pathway. Compellingly, the results show that the protection observed in TRP14 KO mice with pancreatitis is lost when cystathionine β -synthase is inhibited (see on page 13, lines 336-344, as well as the histological analysis in Figure 9E-G and Appendix Figure S4). We thank the Reviewer for this comment, as this additional mouse data clearly further strengthens the study and supports our proposed transsulfuration-dependent mechanisms underlying for the observed protective effects.

9/ I could not find any data availability statement. **All proteomic data must be uploaded to respective databases**

Answer: Thank you for bringing this to our attention – we apologize for not making this more apparent in the previous submission. The proteomic data have been

deposited to PRIDE with accession numbers PXD050210 and PXD050368, as well as in Mendeley data with DOI 10.17632/7h9xb3cm33.1. This is now stated in the revised manuscript in the item on Data availability at the end of Materials and Methods section on pages 33-34.

Minor

1/ **size bar** indicating the dimensions of the microscopic images is missing

Answer: The size bars are now included in the microscopic images in the revised version of the manuscript.

2/ **labelling/font size etc of the images is different** between panels (compare e.g. 1B with 1C); compare Fig. 3 and 4 -AP/CT vs control/pancreatitis?

Answer: Thanks for catching this! The labelling/font is now homogeneous among all panels.

3/ I am surprised that **the authors did not test the susceptibility of the TRP14 KO cells to ferroptosis** (inhibition of system XC- faithfully leads to ferroptosis induction)

Answer: This is of course a very good point, yet the complexities of ferroptosis (cell type-specificities, oncogene associations, etc.) make this a separate and, indeed, a fairly major project. We have preliminary data on the impact of the loss of the TRP14 orthologue in *C. elegans* where ferroptosis promoted by protein aggregation, and we are studying ferroptotic cell death in the HEK293 cell model with or without TRP14, but we consider that these data lay outside of the scope of the present manuscript. Therefore, while we agree with the significance of what the Reviewer has suggested, we hope the Reviewer will appreciate that we feel this would not be appropriate for inclusion in the current manuscript.

4/ **Are their changes in TRP14 levels/activity under disease conditions?**

Answer: TRP14 protein levels did not change significantly in pancreas in acute pancreatitis (see Appendix Figure S3G).

Referee #2:

This is an interesting manuscript proposing that TRP-14 is the major mechanism by which cystine disulfide (cystine) is reduced to the thiol state in cells. The authors also provide data about the **changes of cystine metabolism in WT or TRP-14 null mice during pancreatitis - but I found this confusing** as I will articulate below.

Many of the **Figures are difficult to read with the text being small - well below publication quality** and hinders the review process and the authors communication of their data. There are no Figure numbers.

Answer: Thank you for finding the manuscript to be interesting, and thank you for giving very good comments on how to further improve it. We apologize if the figures weren't optimally made. The Figure numbers have now been included in the revised version of the manuscript, and the composition, size and lettering of all figures has been improved to meet publication quality.

The authors use **BODIPY FL L-Cystine**. In Fig 1A the work use BODIPY FL is used, but this is unhelpful because the crucial fact is that **this relates to cystine** (arguably the most important point) and this is not immediately clear. More importantly, I was surprised the authors were able to use this as a selective substrate for TRP14, which is

what might be taken from Fig 1A as TRP14 KO reduced the signal to essentially zero.

This reagent ([https://urldefense.com/v3/https://www.thermofisher.com/order/catalog/product/B20340_!!D9dNQwwGXtA!Q0vCTzNN7tvihWGVsoF3xSATndJv7KWNHFg_A9tD9XXwObcG1V8T239EdMJv5o-sZZ7sf2dR37KpuYH-5i3XpVSzZY0\\$](https://urldefense.com/v3/https://www.thermofisher.com/order/catalog/product/B20340_!!D9dNQwwGXtA!Q0vCTzNN7tvihWGVsoF3xSATndJv7KWNHFg_A9tD9XXwObcG1V8T239EdMJv5o-sZZ7sf2dR37KpuYH-5i3XpVSzZY0$)) **has been used to index total thiol** because as it says in the manufacturers information, **it thiol-disulfide exchanges to liberate cysteine-FL that fluoresces**. A living cell will be reducing inside, and this will include TRP-15 null cells as having predominantly oxidised thiols is not compatible with life. Therefore, **it is extremely difficult to accept that the null cells cannot reduce this dye**. Of course, the authors may have set a threshold that gives the numbers they report, but the problem is this is not absolutely quantitative and is instead arbitrary units and relative to WT. **This can be done truly quantitatively** (i.e., with proper units on the y axis) **in cell lysates, which is important to address the point that it currently appears that the BODIPY FL L-Cystine is not reducible in the absence of TRP-14. All manner of thiol will reduce it.**

A major limitation is that the authors do not quantify the contribution of TRP-14 to cystine reduction. They quickly move on to Fig 1B with various manipulations before pinning down extremely well that TRP-14 is the principal reducer of cystine. Even in the experiment in Fig 1C showing TRP-14 can reduce cystine, this is not really unexpected. What about other thiol reducing enzymes - they probably do as well in this assay system. **Comparisons with other thiol reducing system, with rates of reduction and enzyme kinetics is warranted given the claims made.**

Answer: Thank you for these well warranted comments. First, many Trx-fold proteins can clearly reduce both BODIPY FL L-Cystine as well as regular cystine in pure enzyme systems, and most likely also in a cellular context. However, what we were studying here is the immediate reduction of the compound as it is taken up from the extracellular space, where we found a notable deficiency in reduction – as visualized by fluorescence from the reduced compound – when cells were depleted of TRP14. This was the initial finding that led us into the whole project described in this study. It is of course absolutely correct that those single time point images could be better solidified by including kinetics of BODIPY FL L-Cystine reduction using WT and KO cells, and using complementation with wildtype and an active site mutant of TRP14, which has now been included, also using longer incubation times that allow for fluorescence to appear in all cells (due to the background activity of BODIPY FL L-Cystine that the Reviewer refers to). These new kinetic assays confirmed the markedly decreased cystine reduction capacity of TRP14 KO cells (Fig. 1C), which was fully recovered when overexpressing a wild-type version of the enzyme in TRP14 KO cells, but not when overexpressing an active site mutant version of the enzyme (Fig. 1D-E). Furthermore, the cystine reduction capacity of parental HEK293 cells overexpressing a wild-type version of TRP14 was higher than that of HEK293 cells overexpressing an active site mutant version of the enzyme (see on page 5, lines 121-127). We have also included statements in the revised paper regarding the cystine reduction capacity of many Trx-fold proteins in vitro, while the cystine reduction capacity in a cellular context is a different case. Together, these new experiments clearly further solidified the paper, so thank you indeed for these comments.

Fig 2A/B - there are no loading controls shown. In **Fig 2B**, I am sure that **any / many thiol reducing system added with decrease the signal and a range of other reducing enzymes would ideally have been compared** and there may have also been a control

with the TRP-14 - as feeding NADPH or NADPH & TRxR may be expected to reduce the signal. **Control for incubation times is missing too. The Prx reactivation experiments in 2E - again, probably any thiol reductant system added to the inactivated Prx will achieve the same.** Perhaps TRP-14 is much 'better' at doing this, but **there needs to be some comparisons and kinetics.**

Answer: Loading controls for this experiment are included in supplementary Figure S3, and control for incubation times are included in the corresponding figures. The efficiencies of the Trx1, glutaredoxin, and glutathione systems in reverting protein-cysteinylation in comparison with TRP14 were tested *in vitro* (as also stated above in the response to the comments by Reviewer 1, and are better discussed in the revision on page 9 and on the first paragraph of page 10, as well as in Figs. 5F-G, and Appendix Fig. S3).

The studies comparing WT with TRP-14 (Txndc17) knock-out mice are interesting, but the choice of performing these experiments is not very clear and the connectivity with the preceding work is questionable, especially as the first part on cystine reduction by TRP-14 is not very compelling. I guess it an attempt to bring a biomedical relevance of the attributed TRP-14 selective reduction of cystine - but is not only premature in my opinion because **the TRP-14 selective reduction of cystine is not robust, but because there is no clear mechanistic explanation for the observations made in the pancreatitis model.** The biological observations in pancreatitis model have intrinsic value, but **mechanistically it does not connect well with the initial focus.** Of course, the measurements of thiol redox state in the tissue relate to the first part of the study, but this is a complex model. **It doesn't make sense to me to try and use this as a demonstration that a failure of TRP-14 to reduce cystine has a biological consequence using a model with such complexity. Why does the null mouse not have elevated cystine basally?**

Answer: Thank you for this comment and for recognizing the intrinsic value of the pancreatitis model! We agree that this is a complex physiological / pathological model and difficult to fully understand, but we do feel the data we present bring mechanistic insights to this important pathology. As mentioned above in the reply to Reviewer 1, in order to better explain our rationale for using the mouse model with pancreatitis, which was indeed based upon its suitability to assess the interplay between TRP14 and the transsulfuration pathway as well as its regulation of protein cysteinylation *in vivo*, in the revised version of the manuscript we state on lines 98-109, that “The interplay revealed here between TRP14 and the transsulfuration pathway as sources of cysteine, shown in metazoan model systems including both nematodes, mice and human, suggests cystine reduction, at least in part, underlies the evolutionary conservation of this odd Trx family member that has yet not been understood. Moreover, the genetic tools and established stress protocols afforded by the *C. elegans* and mouse models allowed *in vivo* assessment of the roles of TRP14 under conditions of proteotoxic stress or acute pancreatitis, respectively. The results show that while TRP14 upholds intracellular cystine reduction and de-cysteinylation of proteins under normal conditions, its knockout in combination with oxidative stress leads to accentuated activation of Nrf2 and upregulation of the transsulfuration pathway that results in an, at first seemingly paradoxical, protection against cell or tissue damage.”

Overall, I am not convinced that that TRP-14 is a highly selective cystine reductase and that this protein has a crucial role to play in the pathogenesis of pancreatitis. There are some notable differences in the responses of the null mouse to the disease

model compared to WT, but **whether this can be attributed to a failure to reduce cystine is not proven** in my opinion and therefore **the observations are really just detailed phenotyping without bringing robust mechanistic understanding**.

Answer: Thank you for these comments. Importantly, we have never stated that TRP14 is a highly selective cystine reductase; what we conclude is that it is the rate limiting cystine reductase in metazoan cells, and that a complementary and also important source of cytosolic cysteine is that derived from transsulfuration. We also emphasize that when TRP14 is knocked out transsulfuration was upregulated, which we suggest is the protective effect in pancreatitis. The new AOAA experiments (Fig. 9E-G and Appendix Figure S4) and the flux experiments in HEK293 cells (Fig. 2) further support this notion. We hope that the additional experiments and further explanations of these conclusions in the revised paper have helped to clarify this mechanism of action, with TRP14 and transsulfuration being functionally linked and both being important sources of cysteine.

Referee #3:

This manuscript addresses the question of what happens to cystine taken up by cells and how it is reduced to cysteine such that it can be fed into biosynthetic processes. This seems to be quite a **fundamental question in cell biology, and the authors provide convincing evidence that TRP14 (TXND17) is responsible**. The experiments are comprehensive, and **the results are convincing**.

Answer: We appreciate very much the positive comments of this reviewer as well as her/his minor corrections.

The comments this reviewer has are minor and mostly textual:

Line 118-119: "We next attempted to block the nematode cystine reduction pathway by generating a *txdc-17*(*syb4767*) null allele by CRISPR-Cas9 technology." It would be better if this sentence could avoid the assumption that *txdc-17* in worms would be on the cystine reduction pathway (and let it emerge as an outcome of the experiments). For example, something like, **"To investigate the role of *C. elegans* TRP14 in vivo, we generated..."**

Answer: We appreciate this suggestion. Accordingly, we have revised this sentence as follows: "To investigate the role of the *C. elegans* TRP14 orthologue *in vivo*, we generated a *txdc-17*(*syb4767*) null allele by CRISPR-Cas9 technology" (see on page 7, lines 172-173).

Line 121: "viable and indistinguishable from wild type"

The authors should state in the main text by what parameters the worms were indistinguishable (growth and viability). Presumably there is some other analysis that could be done that would distinguish the *txdc-17* worms from wild type.

Answer: Regarding this request for parameters of growth and viability to demonstrate that the mutant worms used were viable and indistinguishable from wild type worms, in the revised version of the manuscript we are providing further details on developmental analysis, motility assays as well as determination of the size of the worms at the time they reach adulthood (see Fig. 4C, and Appendix Figure S2), which are the usual

parameters to conclude that they are indistinguishable from wild type controls under basal, non-stressed conditions.

Lines 121-127: The description of the mutants is a bit confusing. It is stated that first the double mutant *cbs-1* and *cbs-2* was made, but the next sentence just reports the growth of *cbs-1*. Also, **it is not clear what is the connection between "Together with the fact that *cbs-2* has been proposed to be a pseudogene" and the decision to impair the transsulfuration pathway at a different level. Even if *cbs-2* were not a pseudogene, it seems that the authors would have had to use another strategy to affect the transsulfuration pathway.**

Answer: We agree our description was confusing and we thank the Reviewer for bringing this to our attention. In the revised version of the manuscript, we provide more detailed descriptions of the *cbs-1* and *cbs-2* mutants as well as the double mutant *cbs-1* and *cbs-2* to better explain our strategy to affect the transsulfuration pathway in the nematode model (see on page 7, third paragraph, and Fig. 4B-D).

Line 152: There should be a comma after "protein cysteinylated" in the subheading.

Answer: Yes, thank you for this editing correction, which has been included in the revised version of the manuscript.

Line 174: This reviewer does not understand **Fig. 2C. How does this heat map show higher cysteinylated levels in TRP14 knockdown cells for all the indicated proteins?**

Answer: We appreciate this justified comment raised by the Reviewer, as the heat map was not well explained in the previous version of the manuscript. Hence, we have included the following explanation in the legend of Figure 6: "The proteomic analysis consisted of three different experiments in which wild-type cells (NC-2) and TRP14 knockdown cells (TRP14-20) were incubated with biotinylated cysteine and then enriched for cysteinylated proteins using a streptavidin column. As a control for the technique, cells incubated with cysteine (no label) were processed in parallel to compensate for any non-specific binding to the column. Once the proteomic data were retrieved, we worked with those proteins that were identified in both wild-type and TRP14 knockout cells in the three independent experiments, but not in the unlabeled samples. The cysteinylated peptides were identified, and the signal intensity of each detected precursor peptide was compared between groups. After performing a t-test, we identified 42 proteins that were significantly more cysteinylated in TRP14 KD cells than in WT cells ($p < 0.05$). These are the proteins displayed in the heatmap, which shows the logarithm of the intensity of the precursor signal for each of the proteins (the darker the blue, the higher the signal). This signal is normalized to the sum of the intensities of all peptides detected in the sample." In some cases, the fold change in cysteinylated levels is much higher (elongation factor 2) than in others (peroxiredoxin 1), but all proteins shown in the figure exhibited significantly increased cysteinylated levels in TRP14 KD cells ($p < 0.05$ vs. WT).

Lines 301-305: This sentence is a bit convoluted.

Answer: Following the Reviewer's suggestion, we have revised these two sentences as follows: "With TRP14 being the major cytosolic reductase of cystine in human, *C. elegans*, and mouse cells, and considering that the transsulfuration pathway is the only

reported alternative source of intracellular cysteine, these two cysteine sources must be functionally coordinated and linked. Consistent with this, deletion of TRP14 has little impact on cell physiology as long as the transsulfuration pathway can provide cells with sufficient cysteine; however TRP14 became essential in *C. elegans* under conditions of proteotoxic stress when transsulfuration was absent” (see lines 373-380).

Line 370: "HEK203" is presumably a typo.

Answer: Yes, it should be “HEK293” and this has been corrected in the revised version of the manuscript.

Lines 423-434: Reference 14 is given for a description of Txndc17 mice, but that reference seems only to describe knockdown in cells. Is reference 15 the correct one?

Answer: Yes, thank you for this correction. Indeed, the reference 15 should be cited in that sentence and this has been corrected in the revised version of the manuscript.

There are **spelling and formatting mistakes** in the methods section. Here are a few examples:

Answer: The method section has been revised thoroughly and in detail to correct the spelling and formatting mistakes.

Line 442: "recrystalized"

Answer: This has been corrected in the revised version of the manuscript.

Line 552: "H2O2"

Answer: This has been corrected in the revised version of the manuscript.

Line 595: "scrapping the cells with a disposable cell scrapper" (it should presumably be "scraping the cells with a disposable cell scraper.")

Answer: This has been corrected in the revised version of the manuscript.

Prof. Juan Sastre
University of Valencia, Faculty of Pharmacy
Department of Physiology
Avda. Vicente Andrés Estellés s/n, School of Pharmacy
Burjasot, Valencia 46100
Spain

15th Apr 2024

Re: EMBOJ-2023-114299R

TRP14 is the rate limiting enzyme for intracellular cystine reduction and regulates proteome cysteinylolation

Dear Drs. Sastre and Arnér,

Thank you for submitting your revised manuscript to The EMBO Journal. It has now been seen once more by two of the original referees, and I am happy to say that there are no more objections towards publication now. After incorporation of the remaining editorial issues listed below, we should therefore be able to proceed with formal acceptance of the study:

- Please carefully review the attached Figure preparation guide (also available from our Guide to Authors website) and rearrange the figures and especially their labeling (as also noted by referee 1) accordingly. If this may make some of the figures too dense or too extensive, please consider moving some of the main figure panels into "Expanded View Figures", of which there can be up to five, and which will be typeset and directly viewable online just like the main figures (for more information, please refer to www.embopress.org/page/journal/14602075/authorguide#expandedview)
- On the abstract page of the manuscript, please include 4-5 general keyword terms to enhance searchability.
- Please rename the conflict of interest statement into "Disclosure and competing interests statement" as specified in our Guide to Authors.
- As we are switching from a free-text author contribution statement towards a more formal statement based on Contributor Role Taxonomy (CRediT) terms, please remove the present Author Contribution section and instead specify each author's contribution(s) directly in the Author Information page of our submission system during upload of the final manuscript. See <https://casrai.org/credit/> for more information.
- Please make sure to consistently refer to Appendix Tables as "Appendix Table S1/2/3..." in the text, the Appendix ToC, and the Appendix legends
- Please double-check to make sure that all relevant funding information in the manuscript is congruent with the info entered into our submission system. (K-22 #143769 seems to be missing in the manuscript file?)
- During our routine pre-acceptance checks, we noted that the tubulin loading control is identical between the "Pancreatitis" samples in Figures 8F and 8K. This needs to be clarified, and (in case that Trx1 and NRF2 have been analyzed on the same blot/in the same experiment) clearly explained in the figure legends.
- Our data editors have raised the following queries regarding figures, data, and legends - please make the required modifications with activated TRACK CHANGES option in the manuscript file to facilitate our checking, and also briefly describe your answers/actions in the resubmission cover letter:
 - * Please note that a separate 'Data Information' section is required in the legends of figures 1b-c, e; 2a-c; 3a, c; 4c-d; 7a-d; 8a-e, g-j; 9b-d, f-g.
 - * Please define the annotated p values ****/****/**/*/ #####/ #####/ {section sign}{section sign}{section sign}{section sign}/ {section sign}{section sign}/ {section sign} in the legend of figure 1b-c, e; 2a-c; 3a, c; 8a-e, g-j; 9b-d, f-g; as appropriate.
 - * Please indicate the statistical test used for data analysis in the legends of figures 1b-c, e; 2a-c; 3a, c; 6; 7a-d; 8a-e, g-j; 9b-d, f-g.
 - * Please note that in figures 7a-d; there is a mismatch between the annotated p values in the figure legend and the annotated p values in the figure file that should be corrected.
 - * Although 'n' is provided, please describe the nature of entity for 'n' in the legends of figures 1b-c, e; 2a-c; 3a, c; 4d; 7a-d; 8a-e, g-j; 9b-d, f-g.
 - * Please note that the error bars are not defined in the legends of figures 1b-c, e; 2a-c; 3a, c; 4d; 7a-d; 8a-e, g-j; 9b-d, f-g."
 - * Please note that the scale bar needs to be defined for figures 9a, e.
 - * Please note that scale bar and its definition are missing for figures 1b; 3a-b."

- Finally, please provide suggestions for a short 'blurb' text prefacing and summing up the conceptual aspect of the study in two sentences (max. 250 characters), followed by 3-5 one-sentence 'bullet points' with brief factual statements of key results of the paper; they will form the basis of an editor-written 'Synopsis' accompanying the online version of the article. Please also upload a synopsis image, which can be used as a "visual title" for the synopsis section of your paper. The image (considerably simpler than the included Figure 10) should be in PNG or JPG format, and please make sure that it appears well in the rather modest dimensions of (exactly) 550 pixels wide and 300-600 pixels high.

I am therefore returning the manuscript to you for a final round of minor revision, to allow you to make these adjustments and clarifications, and to upload all modified files. Once we will have received them, we should hopefully be ready to swiftly proceed with acceptance and production of the manuscript.

Yours sincerely,

Hartmut Vodermaier

9) Digital image enhancement is acceptable practice, as long as it accurately represents the original data and conforms to community standards. If a figure has been subjected to significant electronic manipulation, this must be clearly noted in the figure legend and/or the 'Materials and Methods' section. The editors reserve the right to request original versions of figures and the original images that were used to assemble the figure. Finally, we generally encourage uploading of numerical as well as gel/blot

image source data; for details see: embopress.org/page/journal/14602075/authorguide#sourcedata

At EMBO Press, we ask authors to provide source data for the main manuscript figures. Our source data coordinator will contact you to discuss which figure panels we would need source data for and will also provide you with helpful tips on how to upload and organize the files.

In the interest of ensuring the conceptual advance provided by the work, we recommend submitting a revision within 3 months (14th Jul 2024). Please discuss the revision progress ahead of this time with the editor if you require more time to complete the revisions. Use the link below to submit your revision:

Link Not Available

Referee #1:

The authors addressed my comments in full and experimentally.

One minor comment: I would encourage the authors to increase the font size on their figures as they might be difficult to read when the final pdf is printed.

Referee #2:

The manuscript has been improved. Nevertheless, I think this feels to be several parts that do not fit together especially well, and what we have is several loosely connected findings that individually are not especially strong but have been pieced together to make a vaguely connected story. I see this as a rather broad study, but the mechanistic depth and strengths are limited with many of the original comments not having been fully addressed during the revision. Despite this, of course this work has value and there may be sufficient content and interest here for publication, but that is an editorial decision.

**POINT-BY-POINT RESPONSE to REVIEWERS
of MANUSCRIPT EMBOJ-2023-114299R**

Please carefully review the attached Figure preparation guide (also available from our Guide to Authors website) and rearrange the figures and especially their labeling (as also noted by referee 1) accordingly. If this may make some of the figures too dense or too extensive, please consider moving some of the main figure panels into "Expanded View Figures", of which there can be up to five, and which will be typeset and directly viewable online just like the main figures.

Answer: The size of labels has been increased in all figures and the Table in Figure 4 has been enlarged following the comment raised by referee 1, and the figures have been revised taking into account the Figure preparation guide. Figure 4 has been rearranged and the previous panel B from Figure 4 has been converted into Expanded View Figure 1 (Fig. EV1) in the final version of the manuscript.

On the abstract page of the manuscript, please include 4-5 general keyword terms to enhance searchability.

Answer: We have added five keywords on the abstract page (page 2).

Please rename the conflict of interest statement into "Disclosure and competing interests statement" as specified in our Guide to Authors.

Answer: The conflict of interest statement has now been renamed as "Disclosure and competing interests statement" (page 36, line 972).

As we are switching from a free-text author contribution statement towards a more formal statement based on Contributor Role Taxonomy (CRediT) terms, please remove the present Author Contribution section and instead specify each author's contribution(s) directly in the Author Information page of our submission system during upload of the final manuscript.

Answer: The Author Contribution Section has been removed from the manuscript and instead, said information has been included in the Author Information page of the online submission system.

Please make sure to consistently refer to Appendix Tables as "Appendix Table S1/2/3..." in the text, the Appendix ToC, and the Appendix legends.

Answer: We have revised the reference to the Appendix Tables in the manuscript (see on page 14, lines 357, and 365-367; page 17, line 444; page 20, line 550; page 29, line 796), the Appendix table of contents, and the Appendix legends.

Please double-check to make sure to all relevant funding information in the manuscript is congruent with the info entered into our submission system (K-22 #143769 seems to be missing in the manuscript file?)

Answer: Thank you for spotting this. In the final version of the manuscript, we have reviewed the funding by adding the Hungarian Eötvös Loránd Kutatási Hálózat Foundation (ELKH) and the Hungarian Magyar Tudományos Akadémia (MTA, grant K-22 #143769) to E.E.S. (see on page 35, lines 947-949).

During our routine pre-acceptance checks, we noted that the tubulin loading control is identical between the "Pancreatitis" samples in Figures 8F and 8K. This needs to be clarified, and (in case that Trx1 and NRF2 have been analyzed on the same blot/in the same experiment) clearly explained in the figure legends.

Answer: Yes, this is correct. Trx1 and NRF2 were analyzed on the same blots, and hence in Figures F and K the tubulin loading control was the same for the pancreatitis samples, and we have stated that in the "Data information section" within the figure legend (page 45, lines 1280-1282).

Our data editors have raised the following queries regarding figures, data, and legends - please make the required modifications with activated TRACK CHANGES option in the manuscript file to facilitate our checking, and also briefly describe your answers/actions in the resubmission cover letter:

Answer: We have made changes to the figure legends with the track changes feature. Please, see our detailed point-by-point responses below:

*** Please note that a separate 'Data Information' section is required in the legends of figures 1b-c, e; 2a-c; 3a, c; 4c-d; 7a-d; 8a-e, g-j; 9b-d, f-g.**

Answer: A data information section was added in the figure legends of figures 1, 2, 3, 4, 6, 7, 8, and 9, as requested (see on pages 41-46).

*** Please define the annotated p values ****/**/**/*/ #####/ ###/ {section sign}{section sign}{section sign}{section sign}/ {section sign}{section sign}/ {section sign} in the legend of figure 1b-c, e; 2a-c; 3a, c; 8a-e, g-j; 9b-d, f-g; as appropriate.**

Answer: The annotated P values have been defined in the data information section of the figure legends from figures 1, 2, 3, 8, and 9 (pages 41-42, and 45).

*** Please indicate the statistical test used for data analysis in the legends of figures 1b-c, e; 2a-c; 3a, c; 6; 7a-d; 8a-e, g-j; 9b-d, f-g.**

Answer: The statistical test used in each case was added in the data information section in the figure legends from figures 1, 2, 3, 6, 7, 8, and 9 (pages 40-41, and 44-45).

*** Please note that in figures 7a-d; there is a mismatch between the annotated p values in the figure legend and the annotated p values in the figure file that should be corrected.**

Answer: Thank you for noticing this mistake. The correct P values have been included on the data information section of the figure legend (page 44).

*** Although 'n' is provided, please describe the nature of entity for 'n' in the legends of figures 1b-c, e; 2a- c; 3a, c; 4d; 7a-d; 8a-e, g-j; 9b-d, f-g.**

Answer: Following the suggestion, the nature for "n" has been described in the data information section of figures 1, 2, 3, 4, 6, 7, 8, and 9 (pages 40-46).

*** Please note that the error bars are not defined in the legends of figures 1b-c, e; 2a-c; 3a, c; 4d; 7a-d; 8a- e, g-j; 9b-d, f-g."**

Answer: Bars and error bars are now defined in the data information section of figures 1, 2, 3, 4, 7, 8, and 9 (pages 40-41 and 44-46).

*** Please note that the scale bar needs to be defined for figures 9a, e.**

Answer: The scale bar is now defined in the data information section of the figure legend (page 46).

*** Please note that scale bar and its definition are missing for figures 1b; 3a-b."**

Answer: We have included the corresponding scale bars in Figures 1 and 3, and they are also defined in the data information section of their corresponding figure legends (pages 41-42).

Finally, please provide suggestions for a short 'blurb' text prefacing and summing up the conceptual aspect of the study in two sentences (max. 250 characters), followed by 3-5 one-sentence 'bullet points' with brief factual statements of key results of the paper; they will form the basis of an editor-written 'Synopsis' accompanying the online version of the article. Please also upload a synopsis image, which can be used as a "visual title" for the synopsis section of your paper. The image (considerably simpler than the included Figure 10) should be in PNG or JPG format, and please make sure that it appears well in the rather modest dimensions of (exactly) 550 pixels wide and 300-600 pixels high.

Answer: Following your instructions, we have prepared a short blurb text and a synopsis image that is 550 px wide and 450 pixels high, and both have been uploaded in the online submission of this final version of the manuscript.

In addition, the PRIDE partner repository identifier for the Proteomic analysis of acute pancreatitis in TRP-KO mice has been included on page 32, line 872, as well as on page 34. In the previous version these mass spectrometry proteomics data was deposited in Mendeley.

Prof. Juan Sastre
University of Valencia, Faculty of Pharmacy
Department of Physiology
Avda. Vicente Andrés Estellés s/n, School of Pharmacy
Burjasot, Valencia 46100
Spain

26th Apr 2024

Re: EMBOJ-2023-114299R1
TRP14 is the rate-limiting enzyme for intracellular cystine reduction and regulates proteome cysteinylolation

Dear Prof. Sastre and Arner,

Thank you for submitting your final revised manuscript for our consideration. I am pleased to inform you that we have now accepted it for publication in The EMBO Journal.

Yours sincerely,

Hartmut Vodermaier
